# Auditory cortex anatomy reflects multilingual phonological experience

Olga Kepinska[1,2]*, Josue Dalboni da Rocha[3], Carola Tuerk[4], Alexis Hervais-Adelman[5,6], Florence Bouhali[7], David W Green[8], Cathy J Price[9], Narly Golestani[1,2,4]

[1]Brain and Language Lab, Vienna Cognitive Science Hub, University of Vienna, Vienna, Austria; [2]Department of Behavioral and Cognitive Biology, Faculty of Life Sciences, University of Vienna, Vienna, Austria; [3]Department of Diagnostic Imaging, St Jude Children's Research Hospital, Memphis, United States; [4]Brain and Language Lab, Department of Psychology, Faculty of Psychology and Educational Sciences, University of Geneva, Geneva, Switzerland; [5]Department of Basic Neuroscience, University of Geneva, Geneva, Switzerland; [6]Zurich Linguistics Centre, University of Zurich, Zurich, Switzerland; [7]Aix Marseille University, CNRS, CRPN, Marseille, France; [8]Experimental Psychology, University College London, London, United Kingdom; [9]Wellcome Trust Centre for Neuroimaging, University College London, London, United Kingdom

*For correspondence:
olga.kepinska@univie.ac.at

Competing interest: The authors declare that no competing interests exist.

## eLife Assessment

This report details **convincing** evidence that experience with multilingualism in general, and with larger phonological inventories specifically, is related to differences in the structure of the transverse temporal gyri. The project is notable for using a relatively large sample, and confirming the primary finding in a second sample. The **important** findings strongly point to experience-dependent plasticity related to language experience as a driver of neuroanatomy of the auditory cortex.

**Abstract** This study examines whether auditory cortex anatomy reflects multilingual experience, specifically individuals' phonological repertoire. Using data from over 200 participants exposed to 1–7 languages across 36 languages, we analyzed the role of language experience and typological distances between languages they spoke in shaping neural signatures of multilingualism. Our findings reveal a negative relationship between the thickness of the left and right second transverse temporal gyrus (TTG) and participants' degree of multilingualism. Models incorporating phoneme-level information in the language experience index explained the most variance in TTG thickness, suggesting that a more extensive and more phonologically diverse language experience is associated with thinner cortices in the second TTG. This pattern, consistent across two datasets, supports the idea of experience-driven pruning and neural efficiency. Our findings indicate that experience with typologically distant languages appear to impact the brain differently than those with similar languages. Moreover, they suggest that early auditory regions seem to represent phoneme-level cross-linguistic information, contrary to the most established models of language processing in the brain, which suggest that phonological processing happens in more lateral posterior superior temporal gyrus (STG) and superior temporal sulcus (STS).

**eLife digest** Our brains are dynamic organs that respond to, and are shaped by, our life experiences. Sometimes, this even results in alterations in the shape and size of their various regions.

The auditory cortex is one of the brain areas processing speech. In people who have mastered more than one tongue, it is also tasked with recognising and distinguishing between speech sounds from different languages.

Although previous research suggests that being bilingual can influence brain anatomy, these changes are still poorly understood. How different parts of the auditory cortex are structured, or how they work together has also remained unclear. Kepinska et al. therefore set out to determine if the structure of the brain's auditory cortex is shaped by language experience. To do so, they used an imaging technique known as structural MRI to take detailed pictures of the auditory cortex of over 200 people who each spoke between one and seven languages. In total, 36 different languages were represented across the entire group.

The scans showed that a specific part of the auditory cortex, called the second TTG, was thinner in people who spoke more languages. Further analysis revealed that some of this variation in TTG thickness was associated with the variety of speech sounds present in the languages that participants were familiar with: the more languages someone spoke, and the greater the sound differences between them, the thinner the second TTG. These results suggest that the auditory cortex is shaped by a process called 'experience-driven efficiency'; in other words, the TTG needs less tissue to do its job in people who have more experience in different languages.

Going forward, Kepinska et al. hope that these findings may help refine our understanding of how the brain adapts to language exposure. This, in turn, could improve both educational interventions and clinical therapies, for example to help people with dyslexia or hearing impairments.

## Introduction

Processing of complex auditory stimuli is fundamental to vocal communication. In humans with typical development, auditory processing has been shown to depend on multiple regions within the temporal cortex spreading across the superior and lateral aspects of the superior temporal gyrus (STG). The superior aspect of the STG hosts tonotopically organized primary auditory regions, which have been shown to be located in regions including the first transverse temporal gyrus (TTG), i.e., Heschl's gyrus (HG) (*Dick et al., 2012*; *Moerel et al., 2014*). The planum polare (PP) is immediately anterior to HG, while the planum temporale (PT) borders it posteriorly. Of note, TTG shows high variability in shape and size, some individuals having one single gyrus, others presenting duplications (with a common stem or fully separated) or multiplication of TTG (*Geschwind and Levitsky, 1968*; *Marie et al., 2015*). The superior temporal sulcus (STS) lies directly below the STG and separates it from the middle temporal gyrus. Understanding functional contributions of these different brain areas to processing of sound and vocal stimuli has been advanced by lesion studies (*Hillis et al., 2017*), ablation, and resection case studies (*Hamilton et al., 2021*; *Hullett et al., 2022*), functional magnetic resonance imaging (fMRI) (*Booth et al., 2002*; *Da Costa et al., 2011*; *Humphries et al., 2014*), and intracranial recordings (*Hamilton et al., 2021*; *Mesgarani et al., 2014*). To date, however, no single functional parcellation of the human auditory cortical areas is generally agreed upon (*Hamilton et al., 2021*; *Moerel et al., 2014*). For example, in contrast to hierarchical models of auditory processing (e.g. *Binder et al., 2000*; *Humphries et al., 2014*; *Scott and Johnsrude, 2003*; *Wessinger et al., 2001*), recent proposals of sound processing postulate that processing of speech is distributed, and happens in parallel in different regions of the auditory cortex (*Hamilton et al., 2021*). According to parallel processing models, the posterior STG has been shown to be a crucial and essential locus for language and phonological processing (*Bhaya-Grossman and Chang, 2022*; *Hillis et al., 2017*), and to encode acoustic-articulatory features of speech sounds (*Lakretz et al., 2021*; *Mesgarani et al., 2014*). The primary auditory regions, including HG, have been argued not to be necessary for speech perception (*Hamilton et al., 2021*; *Hullett et al., 2022*).

Structural brain imaging provides another route for gaining insight into the functional roles of cortical subregions, albeit at a different timescale. Quantifying individual differences in behavioral skill and/or experience and relating them to cortical morphology can inform us about the relative

influences of experience-dependent plasticity and of potential predisposition, in different domains. Of note, distinct influences (environmental *versus* genetic) tend to be reflected by different underlying anatomical characteristics of the brain. Indeed, a recent large-scale genome-wide association meta-analysis suggests that cortical surface area is relatively more influenced by genetics and that cortical thickness tends to reflect environmentally driven neuroplasticity (*Grasby et al., 2020*). Such environmental effect on cortical thickness might in turn be tied to microstructural changes to the underlying brain tissue, such as modifications in dendritic length and branching, synaptogenesis or synaptic pruning, growth of capillaries and glia, all previously tied to some kind of environmental enrichment and/or skill learning (see *Lövdén et al., 2013*; *Zatorre et al., 2012*, for overviews). Increased cortical thickness may reflect synaptogenesis and dendritic growth, while cortical thinning observed with MRI may be a result of increased myelination (*Natu et al., 2019*) or synaptic pruning.

In the context of multilingualism, structural brain imaging can offer insights into whether and how the brain accommodates experience with different languages, and into what type of linguistic information is encoded, stored, and processed by particular brain structures. Previous findings point to an influence of experience-dependent plasticity on thickness of regions associated with speech processing, including the left posterior STG and left PT (*Hervais-Adelman et al., 2017*), as well as the STG (*Mårtensson et al., 2012*). Moreover, *Ressel et al., 2012*, associated lifelong bilingualism with increased volume of the HG, a finding interpreted as reflecting experience-related plasticity related to the acquisition of novel phonology.

Languages differ from each other in many ways and to different degrees: language similarities vary per linguistic domain, with distinct typological distances that can be computed between the languages' phonological, lexical, and syntactic systems. The present study investigates whether there are brain structural signatures of multilingualism in the auditory cortex, and whether typological distances between spoken languages further modulate those signatures. Given our focus on auditory brain regions, we quantified how the languages spoken by the participants differed from each other in terms of their phonological systems. Specifically, we investigated whether the neuroanatomical indices describing the auditory cortex regions were related to cross-linguistic phonological information at different levels: acoustic and articulatory feature-level, phoneme-level, or (more abstract) counts of phonological classes.

To investigate the relationship between the individual variability in the morphology of the auditory cortex and variability in language experience, we analyzed the anatomical brain scans of 204 healthy, right-handed participants from the PLORAS database (*Seghier et al., 2016*). The sample was split into two groups according to the date of data acquisition. The main sample included 136 participants speaking between 1 and 7 languages (2.65 languages on average), and representing a relatively wide range of linguistic diversity (34 different languages in total), see *Figure 1* for a visual representation of the sample's language experience. The replication sample included 68 participants speaking to up to 5 languages, 2.44 languages on average. All MRI anatomical images were processed with FreeSurfer's brain structural pipeline (*Fischl et al., 2004*). Multilingualism was operationalized in a continuous way, without artificially dichotomizing the sample into multi-, bi-, and monolinguals (*DeLuca et al., 2019*; *Luk and Bialystok, 2013*). Given the focus on auditory brain regions and on potential associations

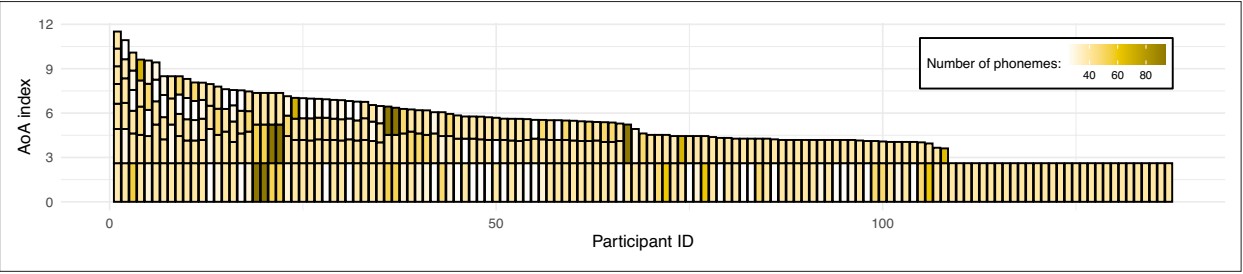

**Figure 1.** Illustration of the sample's language experience. Each bar represents a single participant's overall language experience; the height of the stacked bars within each bar represents the age of onset(s) of acquisition (AoA) index for individual languages (the taller the bar, the earlier in life a given language was acquired). The color of each stacked bar refers to the number of phonemes in each language's phonological inventory. For reference, English phonological inventory has 40 phonemes. Prior to plotting, data was sorted by the overall language experience based on a sum of AoA index for participants' individual languages; consequently, data of participants with most diverse language experience can be found on the left-hand side of the figure, and the right-hand side includes data from monolinguals (i.e. knowing only one language).

with language distance at the phonological level, and the fact that phonological perceptual space is known to be shaped relatively early in life (e.g. *Werker and Hensch, 2015*), we used information regarding the age of onset(s) of acquisition (AoA) of the different languages spoken by participants, and weighed each language by its AoA. For this, we operationalized the language experience in a continuous and quantitative measure for each participant, using *Shannon's, 1948* entropy equation (see Materials and methods, section 'Language experience'), with high entropy values indicating more diverse language experience. In addition to the language experience index not accounting for typological features ('Language experience – no typology'), three measures combining language experience with typological distances at different levels were constructed: (1) 'Language experience – features', (2) 'Language experience – phonemes', and (3) 'Language experience – phonological classes'.

Our analyses aimed at a fine localization of the structural effects of multilingualism. We predicted (see *Bhaya-Grossman and Chang, 2022*; *Hamilton et al., 2021*; *Lakretz et al., 2021*; *Mesgarani et al., 2014*) that the cortical thickness of posterior STG would be related to the composite measures of language experience in our multilingual participants, and to the differences between their languages at the acoustic and articulatory feature level. In terms of the predicted direction of these effects, according to most current views, multilingualism is hypothesized to be a dynamic process reshaping brain structure and inducing both increases (in initial stages) and decreases (in peak efficiency) in brain morphological indices as a function of quantity of the multilingual experience (*Pliatsikas, 2020*). Furthermore, acquisition of multiple languages is thought to have additive effects on brain structure, and to induce cortical and subcortical adaptations in order to accommodate the additional languages (see, e.g., *Hervais-Adelman et al., 2018*). The exact nature of such adaptations for multiple languages of phonologically diverse repertoires on auditory brain regions is to date unknown. A positive relationship between the thickness of auditory regions and the degree of multilingualism (weighted by greater phonological distance) would indicate that the auditory regions need to expand in order to accommodate the variability of phonological input in the environment; a negative one would suggest a specialization of the auditory system in relation to such variability (*Pliatsikas, 2020*).

## Materials and methods

### Participants

#### Main sample

The full sample consisted of *N*=146 participants, with *n*=136 complete cases ($M_{age}$ = 35.79, SD = 15.77; 85 females; 31 monolinguals) exposed to up to 7 languages (2.65 languages on average). *Replication sample*. The replication sample consisted of *N* = 69 participants, with *n* = 68 complete cases ($M_{age}$ = 32.04, SD = 11.68; 38 females; 29 monolinguals) exposed to up to 5 languages (2.44 languages on average). In both cases data were excluded only if any of the dependent or independent variables were missing. *Table 1* breaks down the samples according to how many languages were spoken by which number of participants; further information on participants' language background (languages spoken by each participant) is included in *Appendix 1—table 1*. Study approval was obtained by the Joint Ethics Committee of the Institute of Neurology (University College London, London, UK) and National Hospital for Neurology and Neurosurgery (National Health Service Trust, London, UK) under protocol number 00N032 (Study Title: *The Neural Basis of Language and Object Recognition*).

**Table 1.** Breakdown of participants by number of languages they spoke.

| **Main sample** | | | | | | | |
|---|---|---|---|---|---|---|---|
| Number of languages | 1 | 2 | 3 | 4 | 5 | 6 | 7 |
| Number of participants | 29 | 40 | 36 | 18 | 10 | 2 | 1 |
| **Replication sample** | | | | | | | |
| Number of languages | 1 | 2 | 3 | 4 | 5 | 6 | 7 |
| Number of participants | 30 | 4 | 16 | 14 | 5 | 0 | 0 |

## Language experience

To describe the diverse language background of our participants, we expressed AoA of different languages in a continuous quantitative measure. First, AoA of individual languages was log-transformed to minimize differences between values for languages learned later in life, and the values were inverted to express early AoAs as the highest values; a constant value of 1 was added to the index before log-transformation and after inverting the values, each time to avoid values equal to zero. Next, the AoAs of participants' different languages were combined into one 'language experience' index per participant. It was computed using Shannon's entropy equation (*Shannon's, 1948*):

$$H = - \sum_{i=0}^{n} p_i \log_2 p_i$$

where, $n$ stands for the total number of languages a participant has been exposed to and $p_i$ is the AoA index calculated as above. High entropy values indicated more diverse language experience.

## Typological distance measures

Across the two samples, the participants spoke a total of $N = 36$ different languages ($n = 34$ in the main sample, and $n = 20$ in the replication sample): Arabic, Bengali, Cantonese, Creole (Antiguan), Czech, Danish, Dutch, English, Farsi, Finnish, French, German, Greek, Gujarati, Hakka, Hebrew, Hindi, Hokkien, Hungarian, Irish, Italian, Japanese, Korean, Luxembourgish, Malay, Mandarin, Portuguese, Punjabi, Russian, Sindhi, Spanish, Swahili, Swedish, Swiss German, Urdu, and Vietnamese. The PHOIBLE database (*Moran and McCloy, 2019*) and open-source software (*Dediu and Moisik, 2016*) were used to construct three measures of typological distance between the languages: distances between the distinctive phonological features describing the phonemes of each phonological inventory, distances between the sets of phonemes of the phonological inventories, and similarity in counts of phonological classes that share certain features, as detailed below. *Appendix 1—figure 2* provides a visual representation of the three distance matrices between all the languages. The three distance measures were all related to each other, as evidenced by low to moderate positive correlations between them, see *Table 2*. *Appendix 1—figure 3* shows the distribution of values for each of the distance measures.

### Acoustic and articulatory features

For each documented phonological inventory, PHOIBLE includes their phonemes (see section 'Phonemes' below) and the lower-level distinctive acoustic and articulatory features describing the phonemes. These features offer a qualitative description of differences between discrete sound categories, and are based on articulatory phonology (*Hayes, 2011*). Every phoneme in every language is described by the following 37 binary features: tone, stress, syllabic, short, long, consonantal, sonorant, continuant, delayed release, approximant, tap, trill, nasal, lateral, labial, round, labiodental, coronal, anterior, distributed, strident, dorsal, high, low, front, back, tense, retracted tongue root, advanced tongue root, periodic glottal source, epilaryngeal source, spread glottis, constricted glottis, fortis, raised larynx ejective, lowered larynx implosive, click. Individual languages were described in terms of presence or absence of these features across all phonemes belonging to the given phonological inventory, and represented as vectors. For a given pair of language feature vectors, a pairwise distance was computed as cosine distance using the `scipy.spatial.distance` function in Python.

### Phonemes

Phonemes are individual speech sounds such as vowels and consonants, representing the smallest units of sound that can change the meaning of a word when substituted. Lists of phonemes belonging

**Table 2.** Pearson correlations between the three phonological distance measures.

|  | Features distances | Phoneme distances | Phonological classes distances |
| --- | --- | --- | --- |
| Features distances | 1 |  |  |
| Phoneme distances | 0.42 | 1 |  |
| Phonological classes distances | 0.24 | 0.51 | 1 |

to each of the languages represented in the database were exported from PHOIBLE (*Moran and McCloy, 2019*) and represented as strings. For a given pair of languages *A* and *B*, a pairwise string distance between phonemes of both languages was computed with the Jaccard distance method:

$$1 - J\left(A, B\right) = \frac{|A \cap B|}{|A \cup B|}$$

representing the inverse of the ratio between the size of the intersection and the size of the union of the two sets of phonemes, and varying between 0 (both languages having exactly the same phonemes) and 1 (no phonemes in common).

In addition, using the language-specific phoneme-level information, we constructed a cross-linguistic description of individual languages and used it to compute a measure of cumulative phoneme inventory for each participant. We counted how many unique phonemes each participant was exposed to across all their languages (i.e. phonemes that overlapped between languages were counted only once). For example, the cumulative phoneme inventory of a participant who was exposed only to English was equal to 40, while an English-French bilingual's cumulative phoneme inventory would be equal to 64 (both English and French have 40 unique phonemes [*Stanford Phonology Archive, 2019a*; *Stanford Phonology Archive, 2019b*], with 16 phonemes overlapping across the two inventories).

### Counts of phonological classes

Both the lists of phonemes and phonological features describing them can be used to derive a more abstract, higher-level description of individual languages in terms of which and how many *phonological classes* their phonological inventories contain. Using the PHOIBLE inventories and feature system, *Dediu and Moisik, 2016*, proposed a comprehensive list of 167 phonological classes, along with a method for selecting a language's phonemes belonging to given classes and computing their counts. The system describes how many 'segments', 'consonants', 'vowels', 'diphthongs', etc. a given language has, and counts segments belonging to classes that share certain features (e.g. 'bilabial consonants', 'front vowels', 'clicks', see Appendix 1 –[Phonological classes as defined by *Dediu and Moisik, 2016* for the full list]). Per language, we created a vector representing the counts of all 167 phonological classes, and computed pairwise distances between the languages spoken by the participants in our database with cosine distance (as above).

### Combining typology and language experience (indexed by AoA)

Quantifying diverse language experience with Shannon's entropy equation does not account for similarities and differences between the languages. Therefore, using Rao's quadratic entropy equation (*QE*) (*Bello et al., 2007*; *Pavoine and Bonsall, 2011*; *Rao, 1982*), the language experience index was combined with each of the above typological distance measures, separately, to generate weighted measures accounting for the three different types of cross-linguistic phonological distances. The summed phonological distances (feature-level, phoneme-level, or based on counts of phonological classes) between all language pairs were weighted by the log-transformed and inverted AoA index for each language, as follows:

> If the AoA index for $i$th language in a participant's repertoire of a total of *S* languages is $p_i$ and the dissimilarity between language $i$ and $j$ is $d_{ij}$, then language experience index accounting for typological information has the form:
>
> $$QE = \sum_{i=1}^{S} \sum_{j=1}^{S} d_{ij} p_i p_j$$
>
> where $d_{ij}$ varies from 0 (i.e. where the two languages have exactly the same phonological systems) to 1 (i.e. the two languages have completely different phonological systems). The SYNCSA R package (*Debastiani and Pillar, 2012*; https://rdrr.io/cran/SYNCSA/) was used to perform calculations.

For a similar methodology, but applied to lexical distance measures, see *Kepinska et al., 2023*. In sum, apart from the language experience index not accounting for typological features ('Language experience – no typology'), three measures combining typology and language experience were constructed: (1) 'Language experience – features', (2) 'Language experience – phonemes', and (3) 'Language experience – phonological classes'.

## Neuroimaging data acquisition and processing

For all participants, structural MRIs with 176 sagittal slices were acquired using a T1-weighted three-dimensional (3D) Modified Driven Equilibrium Fourier Transform (MDEFT) sequence. The resulting images had a matrix size of 256×224, yielding a final resolution (voxel size) of 1×1×1 mm$^3$. Repetition time (TR)/echo time (TE)/inversion time (TI) was: [12.24/3.56/530 ms] for the main sample (from a 1.5 T Siemens Sonata scanner), and 7.92/2.48/910 ms for the replication sample (from a 3 T Siemens Trio scanner). The T1 images were denoised using the spatially adaptive non-local means filter (SANLM; *Manjón et al., 2010*) in MATLAB within the CAT12 toolbox. The images were then processed with FreeSurfer's (version 7.2) brain structural pipeline (*Fischl et al., 2004*), which consists of motion correction, intensity normalization, skull stripping, and reconstruction of the volume's voxels into white and pial surfaces.

## Parcellation of the auditory regions

The reconstructed surfaces were parcellated into regions using an atlas-based procedure (*Destrieux et al., 2010*). The auditory regions were delineated using the following labels from the Destrieux atlas: planum polare of the superior temporal gyrus (`G_temp_sup-Plan_polar`; PP), anterior transverse temporal gyrus (of Heschl) (`G_temp_sup-G_T_transv`; referring to HG, or to the anterior-most TTG), transverse temporal sulcus (`S_temporal_transverse`; HS), planum temporale, or temporal plane of the superior temporal gyrus (`G_temp_sup-Plan_tempo`; PT), superior temporal sulcus (parallel sulcus) (`S_temporal_sup`; STS), and lateral aspect of the superior temporal gyrus (`G_temp_sup-Lateral`; STG). The STG and the STS labels were further divided into smaller regions using FreeSurfer's freeview software and FreeSurfer command line tools. For this, on the inflated surface of the fsaverage template, we first drew additional ROIs, dividing the STG into anterior and posterior parts, with the transverse temporal sulcus as the dividing landmark (*Hamilton et al., 2021*). We further divided the STS into anterior, middle, and posterior parts. The anterior versus mid-STS were divided by again aligning the boundary with the posterior border of the anterior TTG, and the mid- versus posterior STS were divided by aligning the boundary with the posterior border of STG. *Appendix 1—figure 4* shows an example of the parcellation in the native space of one of the participants. Next, we transformed the new ROIs into the participants native space (using `mri_surf2surf`), and intersected them with individual subjects' unaltered STG and STS labels. Cortical volume, surface area, and average thickness were computed and extracted from the final labels (with `mris_anatomical_stats`), and were used in the linear mixed models reported in section 'Auditory cortex and language experience'.

For analysis on the TTG along the superior temporal plane (i.e. on the superior aspect of the STG) (see Results, section 'Superior temporal plane and language experience'), we used an automated toolbox (TASH; *Dalboni da Rocha et al., 2020*). TASH runs on the output of the FreeSurfer Destrieux atlas structural segmentation, and provides a finer segmentation not only of HG – whether a single gyrus or a common stem duplication (default version of TASH) – but also of additional TTGs when present (extended version of TASH, called 'TASH_complete'; *Dalboni da Rocha et al., 2023*). Volume, surface area, and average thickness of the resulting labels were extracted. To explore the significant interaction between average thickness of PT and language experience in the exploratory analysis on all auditory regions (section 'Auditory cortex and language experience'), we used 'TASH_complete'; this allowed us to explore possible relationships with additional TTGs, when present. We performed a visual selection of the gyri segmented by TASH_complete, excluding from the analysis gyri that lay along the portion of the superior temporal plane that curved vertically (i.e. within the parietal extension, *Honeycutt et al., 2000*), when present. The volume, surface area, and average thickness of the resulting labels were extracted, and used as dependent variables in the statistical analysis (see Results, see section 'Superior temporal plane and language experience').

## Results

*Table 3* provides an overview of the performed analyses and of results.

**Table 3.** Overview of the performed analyses.

| Analysis | Regions(s) | Measures(s) | Motivation | Results |
|---|---|---|---|---|
| Language experience and auditory regions (section 'Auditory cortex and language experience') | Auditory regions segmented by FreeSurfer (anterior STG, posterior STG, anterior STS, posterior STS, HG, HS, PT, and PP) | Cortical volume, surface area, and thickness | Exploratory first step | Relationship between the average thickness of the PT (bilaterally) and language experience (p=0.01) |
| Language experience and TTG (section 'Superior temporal plane and language experience') | Transverse temporal gyri segmented by TASH | Cortical volume, surface area, and thickness and number of gyri | To elucidate above exploratory results | Relationship between the average thickness of the second TTG (bilaterally) and language experience (p=0.01) |
| Vertex-wise analysis (section 'Superior temporal plane and language experience') | Whole-brain | Cortical thickness | To confirm above exploratory results | Cluster of vertices negatively related to participants' language experience at p<0.0001 (uncorrected), located in the superior aspect of the left STG, corresponding to the location of the second TTG |
| Effects of language proficiency (section 'Superior temporal plane and language experience') | Second TTG | Cortical thickness | To confirm the results obtained from AoA-based language experience index | Relationship between the average thickness of the second TTG (bilaterally) is confirmed using a different metric to calculate language experience (p=0.018 for left, and p=0.065 for right hemisphere) |
| Language typology and second TTG (section 'Second transverse temporal gyrus and effects of language typology') | Second TTG | Cortical thickness | To assess the effect of language typology within the identified region | Quantifying the distance between languages based on phoneme-level information explains the most variance in the average thickness of the bilateral second TTG |
| Language experience expressed as 'cumulative phoneme inventory' (section 'Second transverse temporal gyrus and effects of language typology') | Second TTG | Cortical thickness | To account for language experience in participants with different L1s | Relationship between the thickness of the left second TTG and the specific characteristics of languages at the phoneme-level of their phonological inventories (p=0.004) |
| Language experience in participants with a single TTG (section 'Language experience in participants with a single TTG') | Right HG (first TTG) and right PT | Cortical thickness | To assess the effect of language experience in participants without multiple TTGs | Positive relationship between the thickness of right HG and language experience (p=0.007); no relationship between the thickness of right PT and language experience (p=0.81) |
| Replication analysis (section 'Replication analysis') | Second TTG | Cortical thickness | To replicate the main result (analysis 'Language experience and TTG' and 'Language typology and second TTG') in an independent sample | Relationship between the average thickness of the second TTG and language experience partially replicated (for left hemisphere, p=0.047) |

## Auditory cortex and language experience

In a first, exploratory analysis, we investigated relationships between the composite language experience measure ('Language experience – no typology') and the volume, surface area, and average thickness of the following auditory subregions: STG, STS, HG, HS, PT, and PP, as parcellated by Free-Surfer (*Destrieux et al., 2010*). Given the large size of STG and STS, we refined their segmentations by dividing the STG into anterior and posterior part, and the STS into anterior, middle, and posterior parts (see Materials and methods, section 'Parcellation of the auditory regions' for details). Using the `lme4` R package (*Bates et al., 2015*), we fit three linear mixed models to the extracted anatomical measures (volume, surface area, average thickness), with participants modeled as random effects, language experience, region of interest (ROI), and hemisphere as fixed effects, controlling for the covariates of age, sex and whole-brain volume, area, or mean thickness. Interaction terms for language experience, ROI, and hemisphere were included in the models to determine whether language experience would differentially affect any of the segmented regions. Significance was calculated using the `lmerTest` package (*Kuznetsova et al., 2017*), which applies Satterthwaite's method to estimate degrees of freedom and generate p-values for mixed models. Out of all investigated cortical measures, only average thickness of the PT, bilaterally, was related to participants' language experience at $p=0.01$, see *Table 2*. Specifically, participants with more extensive and diverse language experience had a significantly thinner PT.

## Superior temporal plane and language experience

Given that full TTG duplications, triplications, etc. by definition belong to the PT, and that previous work has shown relationships between TTG duplications patterns and language aptitude (*Turker et al., 2017*), we wanted to better pinpoint the location of the above effects within PT subregions. To this end, and to gain more detailed insight into the localization of the relationship between the thickness of the PT and participants' language experience, we ran three follow-up analyses, an ROI-based one, a vertex-wise based one, as well as an analysis investigating the effect of language proficiency.

First, we used an automatic toolbox (TASH; *Dalboni da Rocha et al., 2020*) to segment gyri along the superior temporal plane (HG, or first TTG, and additional TTG(s), when present), and to extract their cortical thickness, surface area, and volume. Note that FreeSurfer does not segment additional TTGs specifically, but provides measures for the PT as a whole. We included HG in these follow-up analyses (i.e. even though we had included the FreeSurfer HG ROI in the above, broader analyses), given that the default FreeSurfer pipeline is not fine-tuned for the segmentation of this small and variable region, and is thus error-prone for this ROI (*Dalboni da Rocha et al., 2020*). As expected from

**Table 4.** Number of participants, their demographic and language experience characteristics displaying different overall shapes of the transverse temporal gyrus (TTG) (i.e. total number of identified gyri in the left and right hemisphere).
Last column lists whole sample's descriptive statistics.

| | *Left* | | | | *Right* | | | | **Whole sample:** |
|---|---|---|---|---|---|---|---|---|---|
| Total number of gyri: | 1 | 2 | 3 | 4 | 1 | 2 | 3 | 4 | |
| *N* | 6 | 78 | 47 | 5 | 40 | 78 | 18 | 0 | 136 |
| *Male/female* | 0/6 | 28/50 | 23/24 | 3/2 | 10/30 | 33/45 | 11/7 | – | 54/82 |
| *Proportion male/ proportion female* | 0/0.07 | 0.61/0.51 | 0.29/0.42 | 0.05/0.02 | 0.18/0.37 | 0.61/0.55 | 0.20/0.09 | – | |
| *Age (mean)* | 42.95 | 35.95 | 36.29 | 28.48 | 35.12 | 36.49 | 36.63 | – | 36.11 |
| *Language experience – no typology (mean)* | 0.69 | 0.79 | 0.83 | 0.93 | 0.85 | 0.80 | 0.76 | – | 0.807 |
| *Language experience – no typology (min)* | 0.00 | 0.00 | 0.00 | 0.00 | 0.00 | 0.00 | 0.00 | – | 0.000 |
| *Language experience – no typology (max)* | 1.09 | 1.89 | 1.76 | 1.56 | 1.76 | 1.89 | 1.34 | – | 1.893 |

previous work, the TTG showed large individual variability in overall duplication patterns in the sample. Segmentations revealed different numbers of gyri across hemispheres and participants, ranging from a single gyrus to four identified TTGs, see *Table 4*. On average, participants had 2.375 gyri in the left hemisphere ($SD = 0.63$), and 1.84 in the right ($SD = 0.64$). The number of gyri was not related to participants' language experience, either in the left ($\beta = 0.10$, $t = 0.97$, p=0.335), or in the right hemisphere ($\beta = -0.01$, $t = -0.13$, p=0.90), according to a linear model with number of gyri as dependent and language experience as independent variables (controlling for participants' age and sex).

To localize the effect of language experience on the cortical thickness of the segmented gyri, we fit another linear mixed model with participants modeled as random effects, language experience ('Language experience – no typology'), gyrus (first, second, third), and hemisphere as fixed effects, controlling for the covariates of age, sex, and mean thickness (including interaction terms for language experience, gyrus, and hemisphere). We also ran two additional models on the volume and surface area values of the segmented regions, to confirm that the results were specific to the average thickness values, as in the model above (including all auditory regions, see section 'Auditory cortex and language experience'). Out of all investigated cortical measures, only average thickness of the second TTG (bilaterally) was related to participants' language experience at p<0.01, see *Appendix 1—table 3*.

Second, to confirm the above result, we also ran a whole-brain vertex-wise analysis in FreeSurfer. We fit a general linear model to the cortical thickness surface maps of all subjects (smoothed with a 5 mm kernel), testing for a significant effect of the language experience index ('Language experience – no typology') across the whole brain. We found one cluster of vertices negatively related to participants' language experience at p<0.0001 (uncorrected), located in the superior aspect of the left STG, corresponding to the location of the second TTG, see *Appendix 1—figure 5*. At the same threshold, no vertices were found in the right hemisphere, possibly due to greater, known anatomical variability in right compared to left auditory regions (*Dalboni da Rocha et al., 2020*).

We also explored whether the proficiency attained in the spoken languages was related to the thickness of the second TTG. Analyses calculating cumulative language experience based on the participants' proficiency instead of AoA yielded similar results as above, with a significant negative relationship between the proficiency-based cumulative language experience index and the thickness of the left second TTG, and a trend toward significance for the right second TTG (see Appendix 1 – (Effects of language proficiency)).

## Second TTG and effects of language typology

Building on the above result, we investigated whether the thickness of participants' second TTG was related to cross-linguistic phonological information describing the languages they spoke, *above and beyond* being related to mere experience of speaking several languages. The analysis below was therefore performed on a sub-sample of participants who had multiple TTGs ($n$ =130 in the left and $n$ = 96 in the right hemisphere). We constructed three indices of language experience accounting for different typological relations between languages (see *Kepinska et al., 2023*, for a similar approach applied to lexical distances): (1) 'Language experience – features', (2) 'Language experience – phonemes', and (3) 'Language experience – phonological classes'. We used these indices as dependent variables in a set of multiple regression analyses fit to the average thickness values of the left and right second TTG, and performed a model comparison procedure to gauge which of the typological measure was most related to the thickness of the second TTG. Model comparison was performed by computing differences in explained variance ($\Delta R^2$ *Adjusted*) compared to the baseline model (i.e. with language experience without typological information), and using the Bayesian information criterion (BIC) (*Schwarz, 1978*), where the difference between two BICs was converted into a Bayes factor using the below equation, following *Wagenmakers, 2007*:

$$BF_{10} = exp\frac{\Delta BIC_{01}}{2}$$

The model without typological information ('Language experience – no typology') served as baseline for the comparisons. The analyses showed that out of the three language experience indices – (1) 'Language experience – features', (2) 'Language experience – phonemes', and (3) 'Language experience – phonological classes' – the model containing *phoneme* level information explained the

**Table 5.** Left and right second transverse temporal gyri (TTGs) and language experience. Multiple regression model parameters (parameter estimates and standard errors, in brackets; p-values are listed according to the coding presented underneath the table) for the average cortical thickness values of the second TTG, as predicted by the four language experience indices: (1) the cumulative language experience measure not accounting for typology, and cumulative language experience weighted by overlaps between languages at the level of (2) acoustic/articulatory features, (3) phonemes, and (4) counts of phonological classes. Last two rows present model comparison results (additional variance explained and $BF_{10}$ values). NB. All models including typological information were compared against the 'No typology' model.

| Language experience models: | | Left | | | | Right | | | |
|---|---|---|---|---|---|---|---|---|---|
| | | No typology | Features | Phonemes | Phonological classes | No typology | Features | Phonemes | Phonological classes |
| (Intercept) | $\beta$ | 0.00 | −0.01 | 0.00 | −0.01 | 0.01 | 0.01 | 0.01 | 0.01 |
| | SE | (0.03) | (0.03) | (0.03) | (0.03) | (0.03) | (0.04) | (0.03) | (0.03) |
| Language experience | $\beta$ | −0.12** | −0.33 | −0.35** | −0.60* | −0.10+ | −0.25 | −0.26+ | −0.44 |
| | SE | (0.05) | (0.23) | (0.12) | (0.29) | (0.05) | (0.27) | (0.13) | (0.33) |
| Age | $\beta$ | 0.00 | 0.00 | 0.00 | 0.00 | 0.00 | 0.00 | 0.00 | 0.00 |
| | SE | (0.00) | (0.00) | (0.00) | (0.00) | (0.00) | (0.00) | (0.00) | (0.00) |
| Mean thickness (left/right) | $\beta$ | 1.40*** | 1.41*** | 1.37*** | 1.41*** | 0.94* | 0.99* | 0.92* | 0.99** |
| | SE | (0.30) | (0.31) | (0.30) | (0.30) | (0.37) | (0.38) | (0.37) | (0.37) |
| Sex | $\beta$ | 0.00 | 0.01 | 0.00 | 0.01 | −0.01 | −0.01 | −0.01 | 0.00 |
| | SE | (0.05) | (0.05) | (0.05) | (0.05) | (0.05) | (0.05) | (0.05) | (0.05) |
| Num.Obs. | | 130 | 130 | 130 | 130 | 96 | 96 | 96 | 96 |
| $R^2$ | | 0.20 | 0.17 | 0.21 | 0.18 | 0.16 | 0.13 | 0.16 | 0.14 |
| $R^2$ Adj. | | 0.17 | 0.14 | 0.18 | 0.16 | 0.12 | 0.09 | 0.12 | 0.10 |
| AIC | | 18.5 | 23.5 | 16.9 | 21.2 | 11.9 | 14.9 | 11.8 | 14.0 |
| BIC | | 35.7 | 40.7 | 34.1 | 38.4 | 27.3 | 30.3 | 27.2 | 29.4 |
| Log.Lik. | | −3.23 | −5.75 | −2.43 | −4.62 | 0.04 | −1.47 | 0.10 | −0.99 |
| F | | 7.80 | 6.32 | 8.29 | 6.98 | 4.30 | 3.46 | 4.33 | 3.72 |
| RMSE | | 0.25 | 0.25 | 0.25 | 0.25 | 0.24 | 0.25 | 0.24 | 0.24 |
| $\Delta R^2$ Adjusted | | – | −0.03 | 0.01 | −0.02 | – | −0.03 | 0.00 | −0.02 |
| $BF_{10}$ | | – | 0.08 | 2.23 | 0.25 | – | 0.22 | 1.08 | 0.36 |

.p<0.1, +p = 0.05, *p<0.05, **p<0.01, ***p<0.001.

most variance in the average thickness of the bilateral second TTG. It also outperformed the model containing the language experience index not accounting for any typological information ('Language experience – no typology'), see *Table 5*. None of the other phonological distance measures combined with language experience performed better than language experience alone. The direction of the relationship between language experience weighted by typological distance was negative, and had a small effect size ($\beta = -0.35$, $t = -2.96$, p=0.004, $f^2 = 0.07$ and $\beta = -0.26$, $t = -1.98$, p=0.05, $f^2 = 0.04$ for left and right hemisphere, respectively), showing that the more extensive one's language experience, and the more varied at the phoneme level one's languages are, the thinner the second TTG cortex (see *Figure 2*). The statistics for the overall regression models are presented in Appendix 1 – (Second TTG and effects of language typology).

To further establish that our results reflected the relationship between TTG structure and *phonological* diversity specifically (as opposed to language diversity in a more general sense), we derived

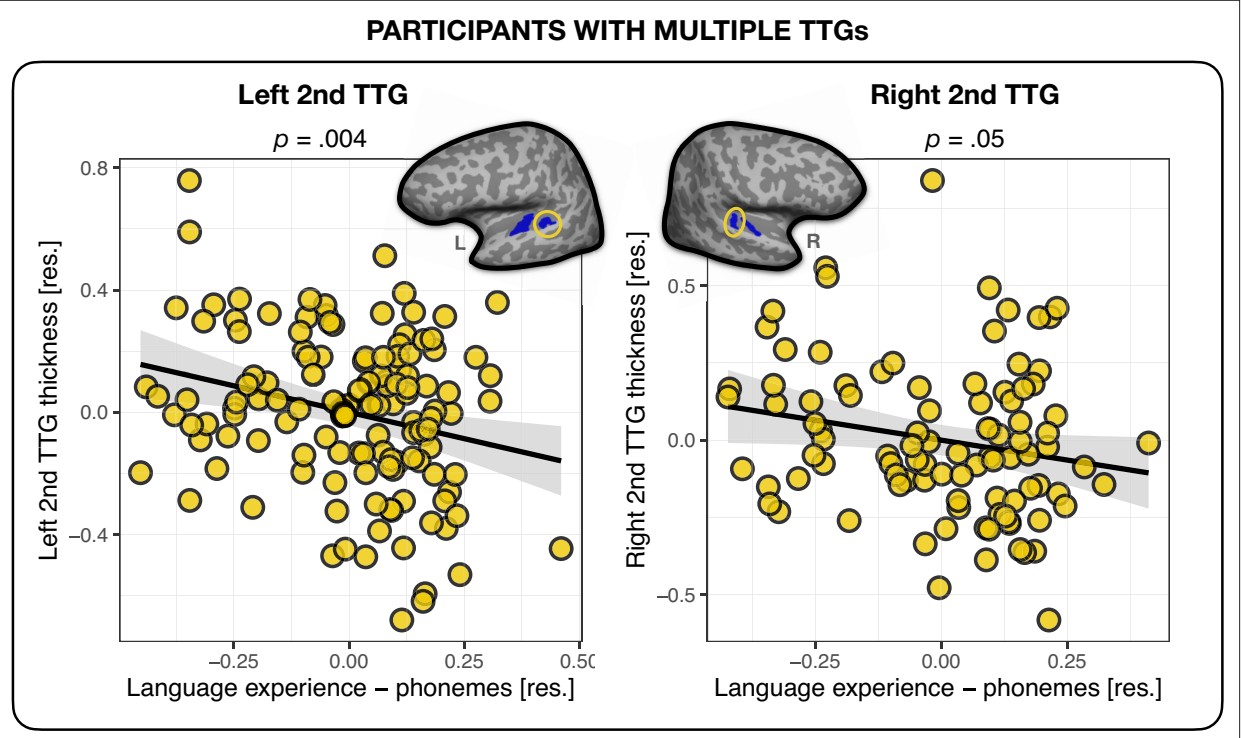

**Figure 2.** Multilingual language experience and thickness of the second transverse temporal gyrus (TTG). Average thickness of the second TTG in the left ($n = 130$) and right ($n = 96$) hemisphere were negatively related to the multilingual language experience index weighted by their phoneme-level phonological distances. Plots show residuals, controlling for age, sex, and mean hemispheric thickness ($\beta = -0.35$, $t = -2.96$, p=0.004 and $\beta = -0.26$, $t = -1.98$, p=0.05 for left and right hemisphere, respectively).

an additional measure of language experience, where the AoA index of different languages was weighted by lexical distances between the languages. Here, we followed the methodology described in *Kepinska et al., 2023*: We used Levenshtein Distance Normalized Divided (LDND) (*Wichmann et al., 2010*) which was computed using the ASJP.R program by Wichmann (https://github.com/Sokiwi/InteractiveASJP01, *Wichmann, 2019*). Information on lexical distances was combined with language experience information per participant using Rao's quadratic entropy equation in the same way as for the phonological measures. We then entered this language experience measure accounting for lexical distances between the languages into linear models predicting the thickness of the second left and right TTG (controlling for participants' age, sex, and mean hemispheric thickness) and observed that the index of multilingual language experience accounting for lexical distances between languages explained less variance than the index incorporating phoneme-level distances between languages, both in the left and the right hemisphere ($R^2_{Adj} = 0.164$ and $R^2_{Adj} = 0.116$ for left and right, respectively). This further strengthens our conclusion that our results reflected the relationship between TTG structure and *phonological* diversity specifically, as opposed to language diversity in a more general sense.

To further probe how cross-linguistic sound inventories may modulate the effect of language experience on the brain, and to account for language experience in participants with different mother tongues (L1s), we constructed a cross-linguistic description of individual languages (see Materials and methods, section 'Phonemes') and used it to calculate a measure of cumulative phoneme inventory for each participant. Here, we counted how many unique phonemes each participant was exposed to across all their languages (i.e. phonemes that overlapped between languages were counted only once), irrespective of when these languages were acquired. We used this measure as an independent variable in another linear model fit to the average thickness values of the second TTG (controlling for age, sex, hemispheric thickness, and accounting for language experience irrespective of language typology) (see section 'Phonemes', Appendix 1– (Cumulative phoneme inventory and the second TTG) and *Appendix 1—table 4*). In the left hemisphere, this measure explained variance in the thickness

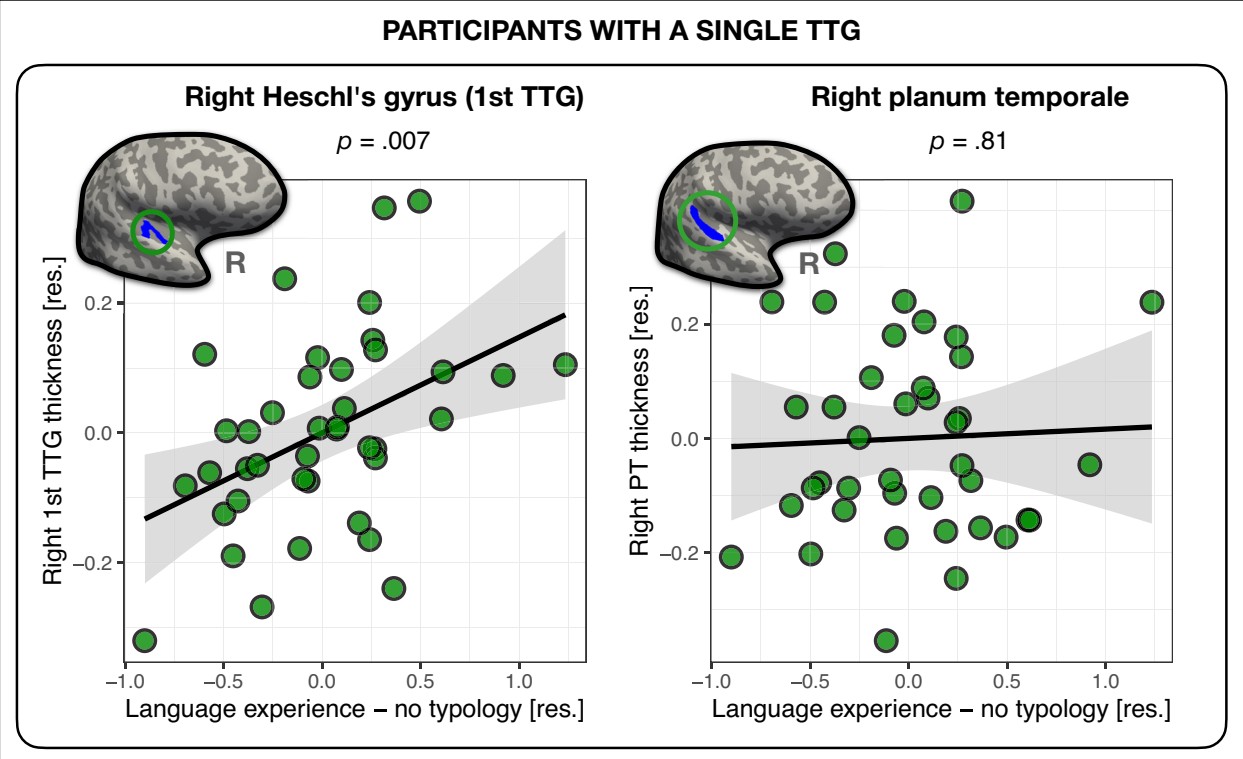

**Figure 3.** Thickness of the right first transverse temporal gyrus (TTG) and planum temporale (PT) in participants with a single TTG in the right hemisphere (*n* = 40). Average thickness of Heschl's gyrus (HG) in the right hemisphere was positively related to the amount of multilingual experience, irrespective of typological relations between languages (*β* = 0.15, *t* = 2.89, p=0.007) (left panel). The average thickness of the right PT was not related to language experience (*β* = 0.02, *t* = 2.43, p=0.81) (right panel). Plots show residuals, controlling for age, sex, and mean hemispheric thickness.

of the second TTG *above and beyond* language experience (*β* = − 0.004, *t* = − 2.910, p=0.004), with a small effect size ($f^2$ = 0.06). In the right hemisphere, the thickness of the second TTG was not significantly related to the 'cumulative phoneme inventory' measure (p=0.76), when also accounting for overall language experience irrespective of typology. This result shows that the thickness of the left second TTG in these participants was related to the specific characteristics of languages at the phoneme level of their phonological inventories, irrespective of when these languages were acquired, while the thickness of the right second TTG was relatively more related to *when and how many* languages were learned (even though the typological distance information did improve the model fit, see *Table 5*), see *Appendix 1—figure 6* and *Appendix 1—table 4*, and Appendix 1 – (Second TTG and effects of language typology).

## Language experience in participants with a single TTG

So far, we revealed associations between phonological language experience and the cortical thickness of the second TTG bilaterally in multilinguals. Yet, 40 participants out of 136 (29.4%) had a single TTG (HG) in the right hemisphere (versus only 6 participants, i.e., 4.4% in the left; see *Table 4*). We therefore repeated the above analysis in the 40 participants with a single TTG in the right hemisphere in order to investigate the relationship between TTG anatomy and language experience in this sample. The analysis revealed that in participants without posterior TTGs in the right hemisphere, the thickness of the first TTG was indeed related to their language experience. The nature of this relation proved however to be different than that for multilingual participants with multiple TTGs. First, the language experience index accounting for most variance in the thickness of the first TTG was the index *not* accounting for typological relations between multilinguals' languages ('Language experience – no typology'), see *Appendix 1—table 5*. Second, the direction of this effect was inverse to that found for participants with multiple TTGs: the more extensive one's language experience, the thicker the HG (*β*=0.15, *t*=2.89, p=0.007; medium effect size: $f^2$=0.237), see *Figure 3*. Of note, we did not find any significant relationship between the thickness of the PT and language experience in this sub-group

of participants ($\beta$=0.02, $t$=2.43, p=0.81). In addition, for PT thickness, compared to a model including only covariates of no-interest, the model including the language experience index offered moderate evidence against it ($BF_{10}$ = 0.16), suggesting that the observed effects between cortical thickness and language experience in the superior temporal plane are specific to gyri only, see *Appendix 1—table 5* and Appendix 1 – (Language experience in participants with a single TTG).

## Replication analysis

In an effort to strengthen the inferences that can be made from the above results, we tested whether the relationship between multilingual language experience and thickness of the second TTG could be replicated in an independent sample of 68 participants. Note that too few participants presented single gyri in the left ($n$ = 7) and the right hemisphere ($n$ = 14), therefore the replication analysis focused on the second TTG results only. Using linear regression, we fit models to the average thickness values of participants' left and right second TTG. As in the analyses on the original sample, we first fit a model using the language experience index irrespective of typology ('Language experience – no typology'), and then we fit models that additionally accounted for the three types of typological relations between languages (features, phonemes, and counts of phonological classes), controlling for covariates of age, sex, scanner model, and mean hemispheric thickness. We again observed that language experience ('Language experience – no typology') did significantly predict the average thickness of the second TTG, but only in the right hemisphere ($\beta$ = – 0.13, $t$ = – 2.05, p=0.046; small effect size: $f^2$ = 0.087), see *Table 6*. Furthermore, similarly to the analysis on the main sample (see section 'Second TTG and effects of language typology'), the language experience index weighted by phonological information at the phoneme level ('Language experience – phonemes') showed a better fit than the model without typological information, explaining 3% of additional variance in the average thickness of the right second TTG. In contrast to the results of the main sample, however, this time the model including phonological feature-level information ('Language experience – features') had a better fit to the thickness data and explained an additional 14% of the variance (11% more variance than the model with language experience weighted by phoneme-level distances), see *Table 6* and *Figure 4C*. No language experience indices were significantly related to the average thickness values of second TTG in the left hemisphere ($\beta$ = – 0.03, $t$ = – 0.53, p=0.60, and $\beta$ = – 0.09, $t$ = – 0.58, p=0.57, for language experience index without and with typological phoneme-level information, respectively), see *Figure 4* and *Table 6*.

## Discussion

Multilingual language experience, quantified in a continuous manner, is related to the structure of the auditory cortex. Using two independent samples of participants, we find replicable effects showing that multilingual language experience, weighed by the age of acquisition of the respective languages, is specifically related to the thickness of gyri (TTG) of the superior temporal plane. Moreover, we show that the typological distances between multilinguals' languages, in particular typological distance measures based on phonemes of the different languages (and to some degree on their acoustic and articulatory features), explains variation in TTG thickness *above and beyond* the effect of AoA alone.

The effects we find are specific to the second TTG in people having at least one full posterior duplication, and to the first TTG in people having only one gyrus. The anterior-most TTG, i.e., HG, is the first cortical relay station for auditory input. HG and the TTG as a whole exhibit large individual variation in shape and size: some individuals have a single TTG, but duplications (complete or partial) and multiplications of the gyrus are also common, and have been observed both ex vivo (*Geschwind and Levitsky, 1968*) and in vivo (*Marie et al., 2015*). The TTG are known to be formed before birth, in utero (*Chi et al., 1977*). Further, genes known to be involved in speech processing, in the development of the nervous system, and in X-linked deafness, have been related to the surface area and thickness of the left and right HG (*Cai et al., 2014*). A body of work has shown relationships between both HG volume and TTG shape (i.e. duplication patterns) and auditory and language abilities that are thought to be relatively stable and thus possibly arising from predisposition, such as speech sound processing and learning abilities (*Golestani et al., 2011*; *Golestani et al., 2007*), linguistic tone learning (*Wong et al., 2008*), and general language aptitude (*Turker et al., 2017*). Studies have also shown larger HG and more TTG duplications in professional musicians (*Schneider et al., 2005*), and longitudinal work

**Table 6.** Thickness of left and right second transverse temporal gyri (TTGs) and language experience in an independent sample of participants.

Multiple regression model parameters (parameter estimates and standard errors, in brackets; p-values are listed according to the coding presented underneath the table) for the average cortical thickness of the second TTG, as predicted by the four language experience indices: (1) the cumulative language experience measure not accounting for typology, and cumulative language experience weighted by overlaps between languages at the level of (2) features, (3) phonemes, and (4) counts of phonological classes. The last two rows present model comparison results (additional variance explained and $BF_{10}$ values). NB. All models including typological information were compared against the 'No typology' model.

| Language experience models: | | *Left* No typology | Features | Phonemes | Phonological classes | *Right* No typology | Features | Phonemes | Phonological classes |
|---|---|---|---|---|---|---|---|---|---|
| (Intercept) | β | 1.00 | 1.12 | 0.99 | 1.24 | –1.85 | –2.20. | –2.09 | –1.21 |
| | SE | (1.14) | (1.10) | (1.13) | (1.09) | (1.34) | (1.21) | (1.33) | (1.30) |
| Language Experience | β | –0.03 | –0.09 | –0.09 | 0.32 | –0.13* | –1.02*** | –0.38* | –0.75 |
| | SE | (0.06) | (0.28) | (0.15) | (0.49) | (0.06) | (0.27) | (0.15) | (0.49) |
| Scanner | β | –0.08 | –0.09 | –0.08 | –0.09 | 0.10 | 0.09 | 0.11 | 0.09 |
| | SE | (0.07) | (0.07) | (0.07) | (0.07) | (0.08) | (0.07) | (0.08) | (0.08) |
| Age | β | 0.00 | 0.01 | 0.00 | 0.01 | 0.01* | 0.01** | 0.01* | 0.01* |
| | SE | (0.00) | (0.00) | (0.00) | (0.00) | (0.00) | (0.00) | (0.00) | (0.00) |
| Sex | β | 0.09 | 0.09 | 0.09 | 0.07 | 0.00 | 0.03 | 0.01 | –0.02 |
| | SE | (0.07) | (0.07) | (0.07) | (0.07) | (0.07) | (0.06) | (0.07) | (0.07) |
| Mean thickness (left/right) | β | 0.57 | 0.52 | 0.58 | 0.42 | 1.72** | 1.86*** | 1.81*** | 1.52** |
| | SE | (0.44) | (0.42) | (0.44) | (0.42) | (0.52) | (0.46) | (0.51) | (0.50) |
| *Num.Obs.* | | 61 | 61 | 61 | 61 | 54 | 54 | 54 | 54 |
| *R²* | | 0.092 | 0.089 | 0.092 | 0.094 | 0.225 | 0.350 | 0.253 | 0.196 |
| *R² Adj.* | | 0.009 | 0.006 | 0.010 | 0.012 | 0.144 | 0.282 | 0.175 | 0.112 |
| *AIC* | | 13.9 | 14.1 | 13.8 | 13.7 | 7.6 | –1.9 | 5.6 | 9.5 |
| *BIC* | | 28.7 | 28.9 | 28.6 | 28.5 | 21.5 | 12.0 | 19.5 | 23.5 |
| *Log.Lik.* | | 0.047 | –0.046 | 0.077 | 0.130 | 3.205 | 7.947 | 4.196 | 2.229 |
| *F* | | 1.109 | 1.072 | 1.120 | 1.142 | 2.781 | 5.158 | 3.244 | 2.342 |
| *RMSE* | | 0.24 | 0.24 | 0.24 | 0.24 | 0.23 | 0.21 | 0.22 | 0.23 |
| *ΔR² Adjusted* | | – | >–0.01 | <0.01 | <0.01 | – | 0.14 | 0.03 | –0.03 |
| *BF₁₀* | | – | 0.91 | 1.03 | 1.09 | – | 114.68 | 2.70 | 0.38 |

.p<0.1, +p = 0.05, *p<0.05, **p<0.01, ***p<0.001.

in children undergoing musical training shows that the gray matter volume of this region is relatively stable over the course of a year (*Seither-Preisler et al., 2014*). To date, however, little evidence has been shown supporting experience-dependent plasticity of TTG. *Ressel et al., 2012*, showed that lifelong, non-elective bilingual language experience with Spanish and Catalan was associated with a larger bilateral HG volume in comparison to monolingual experience. Importantly, the bilinguals in their study were not self-selected, so unlikely to have learned a second language because of a special talent for languages. The findings of the Ressel et al. study thus suggested that the HG volume differences arose from experience with a different phonological system, since Spanish and Catalan are quite similar at the lexical level but have different phonologies. This finding was partially replicated in our

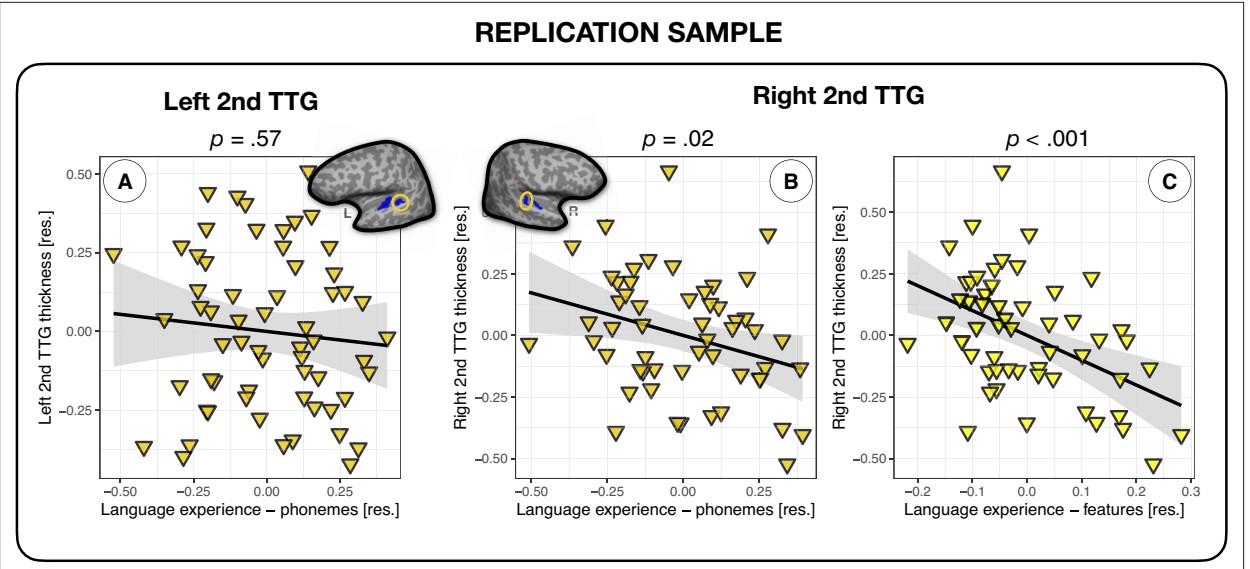

**Figure 4.** Multilingual language experience and thickness of the second transverse temporal gyrus (TTG) in an independent sample of participants. (**A**) Average thickness of the left second TTG ($n$ = 61) was not significantly related to the language experience index ($\beta$ = – 0.09, $t$ = – 0.58, p=0.57); average thickness of the right second TTG ($n$ = 54) was significantly related to the language experience indices accounting for phoneme-level phonological overlaps between multilinguals' languages ($\beta$ = – 0.38, $t$ = – 2.48, p=0.02) (**B**) and feature-level information ($\beta$ = – 1.02, $t$ = – 3.77, p=0.0005) (**C**). The model including phonological feature-level information presented in panel (**C**) had the best fit to the average thickness data of the right second TTG.

study, in a sub-sample of participants with a single gyrus in the right hemisphere. More generally, our results show that experience with multiple languages is associated with differences in the thickness of the TTG, and that increasing degrees of multilingualism are linearly related to the structure of the TTG. Establishing that non-pathological environmental experiences are related to the morphology of TTG offers important insights into the adaptability of this early sensory region, and has important implications for our understanding of the neural underpinnings of auditory processing in general, which we outline below.

We find an effect of multilingualism on the cortical thickness of the second TTG in individuals with a second such gyrus, but on the thickness of the first gyrus in individuals without a full TTG duplication. The result aligns with recent findings of cortical surface area being relatively more associated with genetic factors and of cortical thickness being relatively more associated with environmental factors (e.g. *Eyler et al., 2012*; *Grasby et al., 2020*). Individual differences in the shape (i.e. duplication patterns) of the TTG seem to, however, be related to *how* multilingualism will be accommodated by the thickness of the auditory brain regions: increasing thickness of the first gyrus when no duplications are present, and decreasing thickness of the second gyrus for participants with more than one TTG.

The fact that the most multilingual participants in the current sample had thinner cortex in their second TTG might not automatically point to the conclusion that multilingual language *experience* is associated with this specific neural marker. In fact, polyglotism and multilingualism could partly result from innate predispositions (i.e. language aptitude), which in turn have been associated with brain structural markers within the TTG (*Turker et al., 2017*; *Turker et al., 2021*). However, we observed that accounting for typology (i.e. including information regarding how languages spoken by the participants differed from each other in terms of their phonological systems) explained more variance in the cortical thickness data of the second TTG than an index of language experience accounting for the age of exposure to different languages alone. The contribution of typological distance to our results is important, since it strengthens our interpretation that the observed results are more likely to be due to experience-induced plasticity and not to predispositions: it is far less likely that specific brain anatomical features would induce individuals to learn specific language combinations. It also has to be noted, however, that research has established relationships between population-level genetic factors (distribution of DCDC2 READ1 regulatory element) and characteristics of phoneme inventory of language(s) predominant in that population (*DeMille et al., 2018*; *Tang et al., 2020*),

and that genetic influences on auditory cortex anatomy have previously been observed (*Cai et al., 2014*; *Guadalupe et al., 2014*), making it possible that our results may in some part also be related to genetic rather than purely environmental factors.

Our data show that, within the auditory cortex, language experience is particularly related to gyri in the superior temporal plane, and not to the rest of the PT (as shown in the analysis investigating the sub-sample of participants with a single gyrus in the right hemisphere), nor to other auditory regions such as posterior STG, as shown in the first, exploratory analysis across broader auditory cortex subregions. The posterior STG has been previously shown to be an essential site for language and phonological processing (*Bhaya-Grossman and Chang, 2022*; *Hamilton et al., 2021*; *Hillis et al., 2017*), and for representing acoustic-phonetic features of speech sounds (*Lakretz et al., 2021*; *Mesgarani et al., 2014*). We had predicted that we would find effects of differences between our multilingual participants' languages at the acoustic and articulatory feature level within posterior STG, however indices related to the degree of multilingualism in our sample were not related to its structure. Tuning to the different acoustic-phonetic features might be an experience-independent feature of the posterior STG (i.e. it might be sensitive to features both present and absent from the languages a person knows), and therefore multilingualism might not induce any neuroplastic changes to this region. Alternatively, our findings may suggest that current language models might underestimate the role of early auditory regions (i.e. of HG and second TTG, when present) in phonological processing, and support their role beyond an initial spectro-temporal analysis of speech sounds. Future research, using both structural and functional mapping, is needed to clarify the respective roles of TTG(s) and STG in phonological processing.

With regard to the specific typological features that did best explain our results, we found that accounting for overlaps between languages in terms of their phoneme-level sound structure (i.e. discrete sound categories present in the individual languages' phonological inventories) was the best predictor of the thickness of the second TTG, and that a measure of how many unique phonemes each participant was exposed to across all their languages explained variance in the thickness of the left second TTG *above and beyond* language experience (in the larger main sample). Functional neuroimaging work by *Fisher et al., 2018*, has shown that speech sounds, and specifically vowel formants, are encoded in tonotopic auditory regions including HG and the STG, suggesting that even early auditory regions such as HG and second TTG (when present) are involved in speech sound processing. Similarly, *Bonte et al., 2014* found that vowels can be decoded from fMRI signal of (among others) the temporal plane, HG, and sulcus (see also *Jäncke et al., 2002*; *Obleser and Eisner, 2009*; *van Atteveldt et al., 2004* for findings of processing of isolated phonemes in the HG and PT regions); *Rutten et al., 2019*, showed that attending to speech sounds (stop consonants) in pseudowords results in an increase in the neural processing of higher temporal modulations in regions as early as HG and also in the PT. It is therefore plausible that our finding of an association between greater multilingual language experience and decreased thickness of the second TTG arises from increased processing demands (and associated cortical pruning, see below) inherent to the acquisition and mastery of many non-overlapping phonological inventories. Moreover, to the best of our knowledge, previous studies did not account for fine variation in the exact individual anatomy of the TTG (i.e. single versus multiplicated gyri). Our results may thus call for a re-evaluation of the specific anatomical site within the superior temporal plane in which phonemes are preferentially processed: based on our results, the second TTG (when present) could be a possible candidate.

Apart from the phoneme-based typological distance measure explaining more variance in cortical thickness of the second TTG in the main sample, the data from the replication sample showed that phonological feature-level language distance measure explained additional variance (to the composite language experience measure) in the thickness values. We believe that this inconsistency across samples might be related to the smaller sample size in the replication analysis (and therefore lower power to detect the effect) and the particular phonological characteristics of participants' languages. Specifically, the replication sample included 31 individuals (i.e. 46% of participants) who spoke German and English, a language combination that has a phonological feature distance of zero. In contrast, the main sample included a more heterogenous mix of 34 different languages. Notably, however, in the replication sample, the phonemes-based typological distance measure numerically explained more variance in the thickness of the second TTG than a composite language experience measure based on AoA of different languages, as observed in the main sample.

The decreased thickness of the second TTG in relation to greater multilingualism may arise from different, non-exclusive microstructural and physiological mechanisms, including functional remapping, experience-driven pruning and neural efficiency, or learning-related increased myelination. Primary auditory cortex is known to lie within the anterior-medial part of HG (i.e. the first TTG; *Rademacher et al., 1993*), although there is no exact correspondence between cytoarchitectural and macrostructural boundaries (*Rademacher et al., 2001*). In vivo markers for human primary auditory cortex have been identified, including tonotopy, increased myelination (*Dick et al., 2017*), and decreased cortical thickness (*Zoellner et al., 2019*). Also, there is evidence that in professional musicians, functional activation extends to posterior TTGs during listening to musical sounds, while activation is confined to HG in non-musicians (Schneider et al., unpublished findings). Our findings of thinning in relation to extensive multilingual language experience with a wider diversity of speech sounds could be speculatively interpreted as arising from spatial expansion of primary-like functionality to the second TTG, when present. This spreading of primary-like processing may subsequently induce specialization and pruning of the more posterior gyrus. Alternatively, it could be that the secondary auditory regions beyond HG may be specialized for the processing of (cross-linguistic) phonological information, and this might result in experience-driven pruning. Experience-induced pruning is essential for maintaining an efficient and adaptive neural network. It reinforces relevant neural circuits for faster more efficient information processing, while diminishing those that are less active, or less beneficial. The cortical specialization may need to arise because phonologically more diverse language experience requires that the mapping of acoustic signal to sound categories is denser, more detailed, and more intricate. As a result, the brain may need to engage in more intensive processing to discriminate between and accurately perceive the sound categories of each language. This increased cognitive demand may, in turn, require the auditory and language processing regions of the brain to adapt and become more efficient. Over time, this heightened effort for successful speech perception and sound discrimination may lead to neural plasticity, resulting in cortical specialization. This means that cortical areas become more finely tuned and specialized for processing the unique phonological features of language(s) spoken by individuals. Why this occurs in the second TTG and not in non-gyral portions of the superior temporal plane (i.e. in the PT more generally) remains to be explored in future work, but one possibility is that gyri are more likely to host such a specialized function due to their greater neuronal density or to specific micro-anatomical features and connectivity properties (i.e. greater proximity and thus more direct, local functional, and structural connectivity of neurons within the gyrus, as opposed to with neurons within the sulcal parts). Either way, the neural efficiency account would be in line with the Dynamic Restructuring Model of bilingualism (*Pliatsikas, 2020*), according to which the 'peak efficiency' stage of multilingual functioning comes hand in hand with reductions in local volume. Of note, our supplementary analysis (see Appendix 1 – Effects of language proficiency) investigating second TTG thickness in relation to proficient versus non-proficient languages points to the tentative conclusion that the observed reduced thickness results might be driven by experience with proficiently spoken languages (more so than with non-proficiently spoken ones), aligning with the predictions of the Dynamic Restructuring Model.

Another potential microstructural mechanism underlying the thinner cortex result could be increased myelination of second TTG; this would again be in line with the idea that multilingual language experience leads to modifications such that the second TTG becomes more 'primary-like', since HG is characterized by higher myelination. It has been suggested that the apparent thinning of the ventral temporal cortex during childhood (as argued by, e.g., *Sowell et al., 2004*) is actually driven by increased myelination, since increased myelination in deep cortical layers can shift the apparent gray-white boundary in MR images *Natu et al., 2019*; see also *Mediavilla et al., 2022*, for evidence on learning-related contraction of gray matter being associated with adaptive myelination in rodents. Given the cross-sectional design of the current study, the above explanations remain hypotheses to be tested in further longitudinal studies of language acquisition, preferably ones using myelin mapping (*Lutti et al., 2014*; *Marques et al., 2010*) to explore the underlying physiological mechanisms in more details.

Future research should also further increase the degree of detail in describing the multilingual language experience, as both AoA and proficiency (used here) are not sensitive to other aspects of multilingualism, such as intensity of the exposure to the different languages, or quantity and quality of language input. Since these aspects have been convincingly shown to be associated with neural

changes (e.g. *Romeo, 2019*), incorporating further, more detailed measures describing individuals' language experience could further enhance our understanding of cortical plasticity in general, and how the brain accommodates variable language experience in particular.

Interestingly, in individuals who have only one TTG (i.e. no duplications), we find that multilingualism is related to thickness but in the opposite direction (note that we did not perform a replication analysis of the positive correlation between HG (first TTG) thickness and language experience in the replication sample due to the limited number of individuals presenting with only one gyrus.) We speculate that there might be different mechanisms underlying plasticity in regions more likely to contain primary versus secondary auditory cortex. In other words, it may be more likely that experience-induced pruning occurs in secondary compared to primary regions, and that in primary regions, greater efficiency comes at the cost of a neuronal 'bulking'. Alternately but non-exclusively, it could be that in people who do not have a second TTG, HG needs to accommodate both lower-level auditory processing and also the processing of more complex speech sounds. Further research, especially with larger samples of participants with a single TTG, is needed to elucidate the opposite direction of the relationship between multilingualism and cortical thickness in HG versus the second TTG in our results.

Different methodological tools are available for capturing the variability of TTG. Most previous studies used either manual labeling, whole-brain approaches involving normalizing structural images to a common template space in order to assess regional variation in gray or white matter probability, or automated pipelines serving to segment and label different ROIs (e.g. VBM or FreeSurfer). Both approaches have important advantages, the former allowing for high precision in capturing details of individual variation, the latter two being far less time-consuming and labor-intensive, and therefore more applicable to large datasets. In the current study, we capitalized on recent developments in cortical segmentation efforts and used an automatic toolbox (TASH; *Dalboni da Rocha et al., 2020*) specifically designed for fine delineation of auditory cortex gyri, and extraction of measures describing their anatomy. Coupled with visual inspection of the data, TASH provides a 'best-of-both-worlds' approach, by being anatomically precise while involving little manual involvement with the data (which also tends to be error-prone). The measures we obtained align with the high degree of variability reported in previous literature, but they also seem to be more sensitive, since we identify more multiplications of the TTG than previously reported. For example, in *Marie et al., 2015*, out of 232 subjects, 204 (88%) had one gyrus (including partially duplicated) in the left hemisphere, 28 (12%) had a complete posterior duplication; the prevalence of further multiplications was not assessed in that study. In our data, a minority of participants (4%) presented only one gyrus in the left hemisphere, and most of them had either two (57%) or three (35%) separate gyri. TASH identifies gyri based on anatomical landmarks and curvature values, and unlike previous descriptive work is surface-based rather than volume-based. It therefore might be more sensitive to shallow cortical folds that might otherwise not be discernable in volume-based manual labeling. The precise functional relevance of these multiplications remains to be uncovered in future research, and more work is needed to further ascertain anatomical accuracy of the segmentation of posterior gyri in the superior temporal plane.

Notably, such variability in shape of the auditory regions is absent from other, evolutionarily older, species: macaques' auditory cortex is flat, and some chimpanzees have only a single gyrus (*Hackett et al., 2001*). By showing that a human-specific experience, i.e., speaking multiple languages, is reflected in the anatomy of cortical regions that seem to be evolutionarily particular to humans, we contribute to informing both models of neuroplasticity and of language evolution.

A large body of work has provided insights into both how the brain processes auditory information and speech signals (see *Bhaya-Grossman and Chang, 2022*; *Moerel et al., 2014*, for overviews), and into how bi- and multilingualism are related to the structure of the brain (see *García-Pentón et al., 2016*; *Li et al., 2014*; *Pliatsikas, 2020*, for overviews). In the current study, we brought together both strands of research in an effort to investigate how the auditory cortex accommodates multilingual experience, thereby providing insights into the intricate functioning of auditory processing in humans. Apart from showing that individual anatomy appears to constrain the architecture of phonological representations, our findings also support the idea that experience with typologically similar languages might be different from experience with typologically distant languages (*Antoniou, 2019*; *Berthele, 2020*; *Li et al., 2014*). Indeed, across two independent datasets, we show, for the first time, that cortical thickness of early auditory brain regions is related to

the degree of one's language experience coupled with typological distance between the languages one knows. These findings indicate that early auditory regions seem to represent (or be shaped by) phoneme-level cross-linguistic information, contrary to the most established models of language processing in the brain, which suggest that phonological processing happens in more lateral posterior STG and STS.

The PHOIBLE database (*Moran and McCloy, 2019*) and open-source software (*Dediu and Moisik, 2016*) were used to construct measures of typological distances between the languages. Data were analyzed with publicly available software: CAT12 toolbox (https://neuro-jena.github.io/cat/), Free-Surfer 7.2 (https://surfer.nmr.mgh.harvard.edu), TASH (https://github.com/golestaniBLLab/TASH, *Degano, 2025*), ASJP.R (https://github.com/Sokiwi/InteractiveASJP01, *Wichmann, 2019*), R (with the following packages: stringdist, SYNCSA, entropy, lme4, lmerTest, olsrr, sensemakr) and Python (scipy.spatial.distance function). Figures were created with FreeView (part of FreeSurfer) and R (packages: ggplot2, pheatmap) and compiled in Keynote.

## Acknowledgements

This work was supported by the Wellcome Trust (grant numbers 203147/Z/16/Z and 205103/Z/16/Z) to CJP. The authors gratefully acknowledge support by the NCCR Evolving Language, Swiss National Science Foundation (SNSF) Agreement #51NF40_180888, and by the SNSF grant #100014_182381. OK was funded by the Marie Jahoda Stipendium from University of Vienna. We are grateful to three anonymous reviewers whose feedback greatly improved the manuscript.

## Additional information

### Funding

| Funder | Grant reference number | Author |
| --- | --- | --- |
| Wellcome Trust | 10.35802/203147 | Cathy J Price |
| Wellcome Trust | 10.35802/205103 | Cathy J Price |
| Swiss National Centre of Competence in Research Evolving Language | 51NF40_180888 | Narly Golestani |
| Swiss National Science Foundation | 100014_182381 | Narly Golestani |
| Universität Wien | Marie Jahoda Stipendium | Olga Kepinska |

The funders had no role in study design, data collection and interpretation, or the decision to submit the work for publication. For the purpose of Open Access, the authors have applied a CC BY public copyright license to any Author Accepted Manuscript version arising from this submission.

### Author contributions

Olga Kepinska, Conceptualization, Formal analysis, Investigation, Visualization, Methodology, Writing – original draft, Writing – review and editing; Josue Dalboni da Rocha, Software, Investigation, Methodology, Writing – review and editing; Carola Tuerk, Alexis Hervais-Adelman, Investigation, Writing – review and editing; Florence Bouhali, Formal analysis, Writing – review and editing; David W Green, Data curation, Investigation, Project administration, Writing – review and editing; Cathy J Price, Data curation, Funding acquisition, Investigation, Project administration, Writing – review and editing; Narly Golestani, Conceptualization, Resources, Funding acquisition, Writing – review and editing

### Author ORCIDs

Olga Kepinska  https://orcid.org/0000-0003-2964-1170
Florence Bouhali  https://orcid.org/0000-0001-7357-1531
Narly Golestani  https://orcid.org/0000-0001-7388-4714

## Ethics

All participants gave informed consent to participate in the study, and study approval was obtained by the Joint Ethics Committee of the Institute of Neurology (University College London, London, UK) and National Hospital for Neurology and Neurosurgery (National Health Service Trust, London, UK) under protocol number 00N032 (Study Title: The Neural Basis of Language and object recognition).

Reviewer #1 (Public review): https://doi.org/10.7554/eLife.90269.3.sa1
Reviewer #2 (Public review): https://doi.org/10.7554/eLife.90269.3.sa2
Reviewer #3 (Public review): https://doi.org/10.7554/eLife.90269.3.sa3
Author response https://doi.org/10.7554/eLife.90269.3.sa4

# Additional files

## Supplementary files
MDAR checklist

## Data availability

The current paper used anatomical brain scans of 204 healthy, right-handed participants from the Predicting Language Outcome and Recovery After Stroke (PLORAS) database (*Seghier et al., 2016*). Raw data cannot be made publicly available due to restrictions of the project's ethical approval; raw data can be shared for non-commercial use after contacting the corresponding author and completing a data sharing agreement. Code used for performing the analyses can be accessed through OSF at: https://osf.io/uc52f/ (doi: https://doi.org/10.17605/OSF.IO/UC52F). The OSF repository (https://osf.io/uc52f/) includes all processed data used for the analyses (see under data/data.csv).

The following dataset was generated:

| Author(s) | Year | Dataset title | Dataset URL | Database and Identifier |
|---|---|---|---|---|
| Kepinska O, Dalboni da Rocha JL, Bouhali F, Golestani N | 2025 | Auditory cortex anatomy and multilingualism | https://doi.org/10.17605/OSF.IO/UC52F | Open Science Framework, 10.17605/OSF.IO/UC52F |

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

# Appendix 1

## 1 Participants

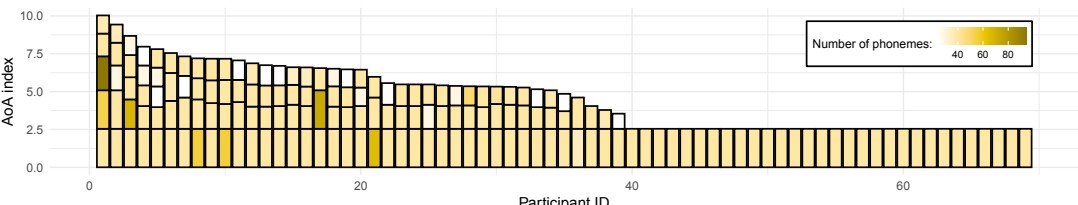

**Appendix 1—figure 1.** Illustration of the replication sample's language experience. As in **Figure 1**, each bar here represents a single participant's overall language experience; the height of the stacked bars within each bar represents the age of onset(s) of acquisition (AoA) index for individual languages (the taller the bar, the earlier in life a given language was acquired). The color of each stacked bar refers to the number of phonemes in each language's phonological inventory.

## 2 Typological distances between languages in the study

### TYPOLOGICAL DISTANCES BETWEEN LANGUAGES IN THE STUDY

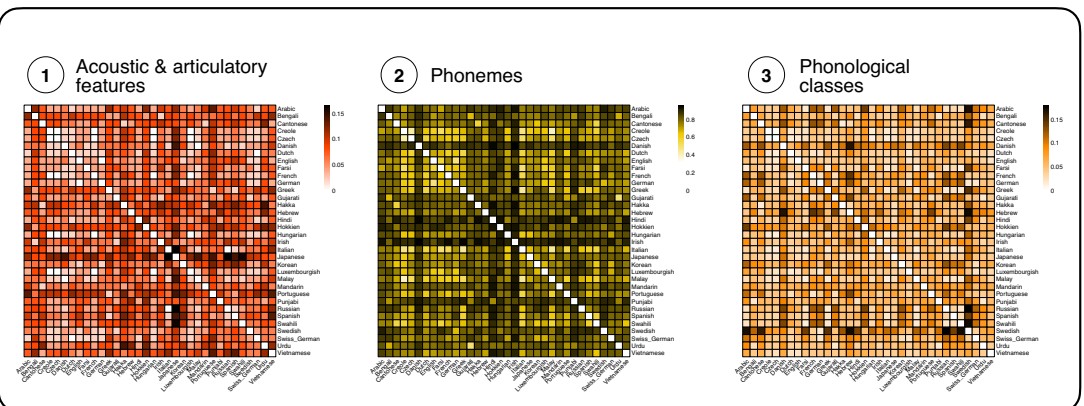

**Appendix 1—figure 2.** Similarity matrices of typological distances between all languages represented in the study (*N* = 36) based on: (1) distances in distinctive acoustic and articulatory features describing the phonemes of each language (e.g. 'short', 'long'); (2) distances in sets of phonemes belonging to each language; and (3) distances based on counts of phonological classes that share certain features (e.g. 'consonants', 'front rounded vowels', 'clicks'). Data for individual languages were collected from the PHOIBLE database (***Stanford Phonology Archive, 2019a***) and open-source software (***Dediu and Moisik, 2016***). The figure was generated in R, with the package pheatmap (***Kolde, 2019***), version 1.0.12.

**Appendix 1—table 1.** Languages spoken by each participant in the main and replication samples.

| Main sample | L1 | L2 | L3 | L4 | L5 | L6 | L7 | Number of languages |
|---|---|---|---|---|---|---|---|---|
| 1 | Czech | Russian | German | Korean | Vietnamese | French | English | 7 |
| 2 | Swiss_German | German | French | Greek | Italian | English | | 6 |
| 3 | German | Italian | English | French | Portuguese | Spanish | | 6 |
| 4 | English | French | German | Spanish | Mandarin | | | 5 |
| 5 | German | English | French | Dutch | Italian | | | 5 |
| 6 | English | French | German | Italian | Spanish | | | 5 |
| 7 | German | English | French | Spanish | Russian | | | 5 |
| 8 | English | German | Luxembourgish | French | Danish | | | 5 |
| 9 | English | German | French | Spanish | Italian | | | 5 |
| 10 | English | Italian | French | German | Spanish | | | 5 |
| 11 | Portuguese | English | Spanish | French | Italian | | | 5 |
| 12 | Hakka | Mandarin | Malay | English | Cantonese | | | 5 |
| 13 | English | French | Italian | Spanish | Mandarin | | | 5 |
| 14 | German | English | French | Japanese | | | | 4 |
| 15 | English | French | German | Danish | | | | 4 |
| 16 | German | English | French | Spanish | | | | 4 |
| 17 | German | English | French | Italian | | | | 4 |
| 18 | German | English | Italian | French | | | | 4 |
| 19 | English | French | German | Italian | | | | 4 |
| 20 | German | English | Greek | French | | | | 4 |
| 21 | French | German | English | Spanish | | | | 4 |
| 22 | Spanish | Hebrew | English | French | | | | 4 |

*Appendix 1—table 1 continued on next page*

Appendix 1—table 1 continued

| Main sample | L1 | L2 | L3 | L4 | L5 | L6 | L7 | Number of languages |
|---|---|---|---|---|---|---|---|---|
| 23 | English | German | Spanish | French | | | | 4 |
| 24 | German | English | French | Italian | | | | 4 |
| 25 | Portuguese | English | French | Italian | | | | 4 |
| 26 | Greek | English | French | Spanish | | | | 4 |
| 27 | Greek | English | French | Spanish | | | | 4 |
| 28 | Greek | English | Italian | Spanish | | | | 4 |
| 29 | Hakka | English | Mandarin | German | | | | 4 |
| 30 | Cantonese | English | Mandarin | Italian | | | | 4 |
| 31 | Cantonese | English | Mandarin | French | | | | 4 |
| 32 | German | English | Dutch | | | | | 3 |
| 33 | English | Spanish | Portuguese | | | | | 3 |
| 34 | English | French | Swedish | | | | | 3 |
| 35 | English | Hindi | Portuguese | | | | | 3 |
| 36 | German | English | French | | | | | 3 |
| 37 | German | English | Dutch | | | | | 3 |
| 38 | German | English | French | | | | | 3 |
| 39 | German | English | Spanish | | | | | 3 |
| 40 | Greek | English | French | | | | | 3 |
| 41 | Greek | English | Spanish | | | | | 3 |
| 42 | Greek | English | German | | | | | 3 |
| 43 | Greek | English | German | | | | | 3 |
| 44 | Greek | English | French | | | | | 3 |

*Appendix 1—table 1 continued*

| Main sample | L1 | L2 | L3 | L4 | L5 | L6 | L7 | Number of languages |
|---|---|---|---|---|---|---|---|---|
| 45 | Greek | English | French | | | | | 3 |
| 46 | Greek | English | French | | | | | 3 |
| 47 | Greek | English | French | | | | | 3 |
| 48 | Greek | English | Italian | | | | | 3 |
| 49 | Mandarin | Cantonese | English | | | | | 3 |
| 50 | Cantonese | English | Mandarin | | | | | 3 |
| 51 | Cantonese | Mandarin | English | | | | | 3 |
| 52 | Cantonese | Mandarin | English | | | | | 3 |
| 53 | English | French | Mandarin | | | | | 3 |
| 54 | Cantonese | English | Mandarin | | | | | 3 |
| 55 | English | Mandarin | German | | | | | 3 |
| 56 | Cantonese | English | Mandarin | | | | | 3 |
| 57 | English | French | Mandarin | | | | | 3 |
| 58 | English | French | Mandarin | | | | | 3 |
| 59 | English | German | Mandarin | | | | | 3 |
| 60 | Bengali | Hindi | English | | | | | 3 |
| 61 | Bengali | Hindi | English | | | | | 3 |
| 62 | Hindi | English | Urdu | | | | | 3 |
| 63 | Hindi | English | Urdu | | | | | 3 |
| 64 | Bengali | English | Hindi | | | | | 3 |
| 65 | Bengali | English | Hindi | | | | | 3 |
| 66 | English | Spanish | French | | | | | 3 |

*Appendix 1—table 1 continued*

| Main sample | L1 | L2 | L3 | L4 | L5 | L6 | L7 | Number of languages |
|---|---|---|---|---|---|---|---|---|
| 67 | Farsi | English | Arabic | | | | | 3 |
| 68 | English | German | | | | | | 2 |
| 69 | English | French | | | | | | 2 |
| 70 | Dutch | English | | | | | | 2 |
| 71 | English | Creole | | | | | | 2 |
| 72 | English | Irish | | | | | | 2 |
| 73 | German | English | | | | | | 2 |
| 74 | German | English | | | | | | 2 |
| 75 | German | English | | | | | | 2 |
| 76 | German | English | | | | | | 2 |
| 77 | English | German | | | | | | 2 |
| 78 | German | English | | | | | | 2 |
| 79 | Italian | English | | | | | | 2 |
| 80 | Hebrew | English | | | | | | 2 |
| 81 | English | Hebrew | | | | | | 2 |
| 82 | Spanish | English | | | | | | 2 |
| 83 | German | English | | | | | | 2 |
| 84 | German | English | | | | | | 2 |
| 85 | Greek | English | | | | | | 2 |
| 86 | Greek | English | | | | | | 2 |
| 87 | Greek | English | | | | | | 2 |
| 88 | Greek | English | | | | | | 2 |

*Appendix 1—table 1 continued*

| Main sample | L1 | L2 | L3 | L4 | L5 | L6 | L7 | Number of languages |
|---|---|---|---|---|---|---|---|---|
| 89 | Greek | English | | | | | | 2 |
| 90 | Greek | English | | | | | | 2 |
| 91 | Greek | English | | | | | | 2 |
| 92 | Greek | English | | | | | | 2 |
| 93 | Mandarin | English | | | | | | 2 |
| 94 | Cantonese | English | | | | | | 2 |
| 95 | Cantonese | English | | | | | | 2 |
| 96 | English | Mandarin | | | | | | 2 |
| 97 | Mandarin | English | | | | | | 2 |
| 98 | Cantonese | English | | | | | | 2 |
| 99 | Mandarin | English | | | | | | 2 |
| 100 | Mandarin | English | | | | | | 2 |
| 101 | Mandarin | English | | | | | | 2 |
| 102 | Gujarati | English | | | | | | 2 |
| 103 | Gujarati | English | | | | | | 2 |
| 104 | Gujarati | English | | | | | | 2 |
| 105 | English | Hungarian | | | | | | 2 |
| 106 | English | French | | | | | | 2 |
| 107 | English | French | | | | | | 2 |
| 108 | English | | | | | | | 1 |
| 109 | English | | | | | | | 1 |
| 110 | English | | | | | | | 1 |

*Appendix 1—table 1 continued on next page*

*Appendix 1—table 1 continued*

| Main sample | L1 | L2 | L3 | L4 | L5 | L6 | L7 | Number of languages |
|---|---|---|---|---|---|---|---|---|
| 111 | English | | | | | | | 1 |
| 112 | English | | | | | | | 1 |
| 113 | English | | | | | | | 1 |
| 114 | English | | | | | | | 1 |
| 115 | English | | | | | | | 1 |
| 116 | English | | | | | | | 1 |
| 117 | English | | | | | | | 1 |
| 118 | English | | | | | | | 1 |
| 119 | English | | | | | | | 1 |
| 120 | English | | | | | | | 1 |
| 121 | English | | | | | | | 1 |
| 122 | English | | | | | | | 1 |
| 123 | English | | | | | | | 1 |
| 124 | English | | | | | | | 1 |
| 125 | English | | | | | | | 1 |
| 126 | English | | | | | | | 1 |
| 127 | English | | | | | | | 1 |
| 128 | English | | | | | | | 1 |
| 129 | English | | | | | | | 1 |
| 130 | English | | | | | | | 1 |
| 131 | English | | | | | | | 1 |
| 132 | English | | | | | | | 1 |

*Appendix 1—table 1 continued on next page*

*Appendix 1—table 1 continued*

**Main sample**

| Main sample | L1 | L2 | L3 | L4 | L5 | L6 | L7 | Number of languages |
|---|---|---|---|---|---|---|---|---|
| 133 | English | | | | | | | 1 |
| 134 | English | | | | | | | 1 |
| 135 | English | | | | | | | 1 |
| 136 | English | | | | | | | 1 |

| Replication sample | L1 | L2 | L3 | L4 | L5 | L6 | L7 | Number of languages |
|---|---|---|---|---|---|---|---|---|
| 1 | German | English | French | Spanish | Italian | | | 5 |
| 2 | German | English | Italian | French | Arabic | | | 5 |
| 3 | Sindhi | English | Hindi | French | Russian | | | 5 |
| 4 | English | Irish | French | German | Italian | | | 5 |
| 5 | German | English | French | Italian | Spanish | | | 5 |
| 6 | English | French | German | Russian | | | | 4 |
| 7 | German | English | Italian | French | | | | 4 |
| 8 | German | English | French | Italian | | | | 4 |
| 9 | English | French | German | Spanish | | | | 4 |
| 10 | English | French | Spanish | German | | | | 4 |
| 11 | Swiss_German | German | English | French | | | | 4 |
| 12 | German | English | French | Spanish | | | | 4 |
| 13 | German | English | French | Swahili | | | | 4 |
| 14 | German | English | French | Swedish | | | | 4 |
| 15 | German | English | French | Spanish | | | | 4 |
| 16 | German | Portuguese | English | French | | | | 4 |
| 17 | German | Dutch | English | French | | | | 4 |

*Appendix 1—table 1 continued*

| Replication sample | L1 | L2 | L3 | L4 | L5 | L6 | L7 | Number of languages |
|---|---|---|---|---|---|---|---|---|
| 18 | Swiss_German | German | French | English | | | | 4 |
| 19 | German | English | French | Spanish | | | | 4 |
| 20 | Hungarian | English | French | | | | | 3 |
| 21 | German | English | French | | | | | 3 |
| 22 | English | German | Spanish | | | | | 3 |
| 23 | Punjabi | English | French | | | | | 3 |
| 24 | English | French | German | | | | | 3 |
| 25 | English | French | German | | | | | 3 |
| 26 | English | French | Italian | | | | | 3 |
| 27 | English | French | Mandarin | | | | | 3 |
| 28 | German | Italian | English | | | | | 3 |
| 29 | German | English | Finnish | | | | | 3 |
| 30 | German | English | French | | | | | 3 |
| 31 | German | English | Spanish | | | | | 3 |
| 32 | German | English | Dutch | | | | | 3 |
| 33 | German | English | Dutch | | | | | 3 |
| 34 | German | English | Spanish | | | | | 3 |
| 35 | German | Russian | English | | | | | 3 |
| 36 | Swahili | English | | | | | | 2 |
| 37 | English | Czech | | | | | | 2 |
| 38 | German | English | | | | | | 2 |
| 39 | English | Spanish | | | | | | 2 |

*Appendix 1—table 1 continued on next page*

*Appendix 1—table 1 continued*

| Replication sample | L1 | L2 | L3 | L4 | L5 | L6 | L7 | Number of languages |
|---|---|---|---|---|---|---|---|---|
| 40 | English | | | | | | | 1 |
| 41 | English | | | | | | | 1 |
| 42 | English | | | | | | | 1 |
| 43 | English | | | | | | | 1 |
| 44 | English | | | | | | | 1 |
| 45 | English | | | | | | | 1 |
| 46 | English | | | | | | | 1 |
| 47 | English | | | | | | | 1 |
| 48 | English | | | | | | | 1 |
| 49 | English | | | | | | | 1 |
| 50 | English | | | | | | | 1 |
| 51 | English | | | | | | | 1 |
| 52 | English | | | | | | | 1 |
| 53 | English | | | | | | | 1 |
| 54 | English | | | | | | | 1 |
| 55 | English | | | | | | | 1 |
| 56 | English | | | | | | | 1 |
| 57 | English | | | | | | | 1 |
| 58 | English | | | | | | | 1 |
| 59 | English | | | | | | | 1 |
| 60 | English | | | | | | | 1 |
| 61 | English | | | | | | | 1 |

*Appendix 1—table 1 continued on next page*

*Appendix 1—table 1 continued*

| Replication sample | L1 | L2 | L3 | L4 | L5 | L6 | L7 | Number of languages |
|---|---|---|---|---|---|---|---|---|
| 62 | English | | | | | | | 1 |
| 63 | English | | | | | | | 1 |
| 64 | English | | | | | | | 1 |
| 65 | English | | | | | | | 1 |
| 66 | English | | | | | | | 1 |
| 67 | English | | | | | | | 1 |
| 68 | English | | | | | | | 1 |
| 69 | English | | | | | | | 1 |

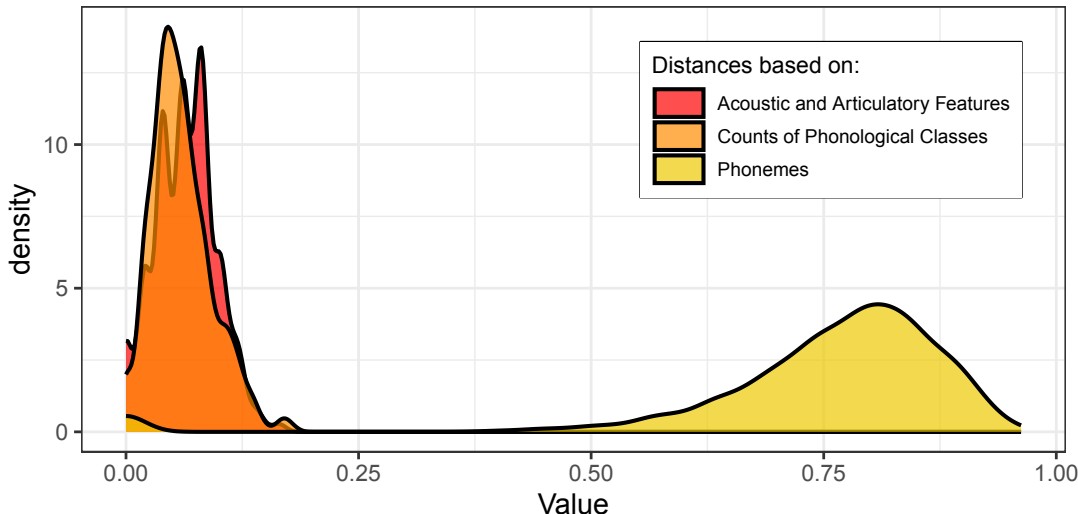

**Appendix 1—figure 3.** Distributions of the three distance measures used in the study based on: (1) distances in distinctive acoustic and articulatory features describing the phonemes of each language (in red); (2) distances in sets of phonemes belonging to each language (in yellow); and (3) distances based on counts of phonological classes (in orange).

## 3 Auditory ROIs used in the analysis

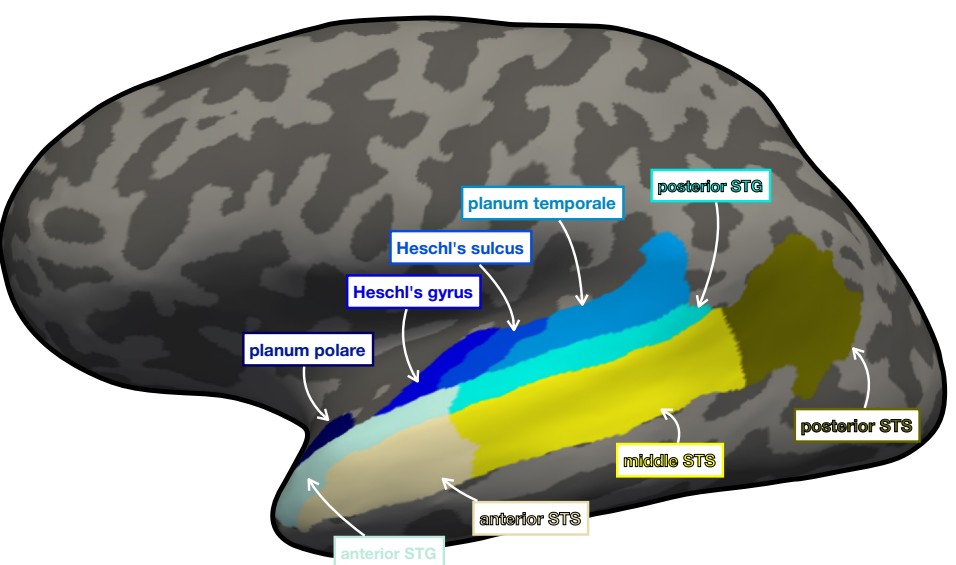

**Appendix 1—figure 4.** Auditory regions of interest (ROIs) used in the analysis. The ROIs are overlaid on an inflated surface in the native space of one of the participants.

## 4 Auditory cortex regions and language experience

**Appendix 1—table 2.** Results of linear mixed models (parameter estimates and standard errors, in brackets; p-values are listed according to the coding presented underneath the table) testing the effect of language experience on the structure (volume, area, and average thickness) of the auditory regions: planum polare, Heschl's gyrus, Heschl's sulcus, planum temporale, anterior and posterior superior temporal gyrus, and anterior, middle, and posterior superior temporal sulcus (STG).

Anterior STG was used as the reference level.

| | | Volume | Area | Thickness |
|---|---|---|---|---|
| (Intercept) | β | 1711.89*** | 58.10*** | 0.58*** |
| | SE | (35.28) | (11.85) | (0.02) |
| Age | β | 1.01 | 0.07 | 0.00 |
| | SE | (0.88) | (0.24) | (0.00) |
| Sex | β | 0.63 | −7.41 | 0.02 |
| | SE | (26.07) | (8.07) | (0.01) |
| Whole-brain: volume/area/thickness | β | 0.00*** | 0.01*** | 0.77*** |
| | SE | (0.00) | (0.00) | (0.09) |
| Language experience | β | 41.50 | 7.45 | 0.03 |
| | SE | (66.09) | (22.33) | (0.03) |
| Posterior STG | β | −1548.51*** | −257.01*** | −0.34*** |
| | SE | (46.38) | (15.85) | (0.02) |
| Anterior STS | β | −2465.10*** | −190.57*** | −0.77*** |
| | SE | (46.38) | (15.85) | (0.02) |
| Middle STS | β | −688.17*** | 646.88*** | −1.01*** |
| | SE | (46.38) | (15.85) | (0.02) |
| Posterior STS | β | −185.92*** | 861.89*** | −1.01*** |
| | SE | (46.38) | (15.85) | (0.02) |
| PT | β | −2064.07*** | −163.35*** | −0.80*** |
| | SE | (46.38) | (15.85) | (0.02) |
| HG | β | −2886.10*** | −482.40*** | −0.78*** |
| | SE | (46.38) | (15.85) | (0.02) |
| HS | β | −3416.84*** | −552.82*** | −0.97*** |
| | SE | (46.38) | (15.85) | (0.02) |
| PP | β | −1870.15*** | −316.73*** | 0.19*** |
| | SE | (46.38) | (15.85) | (0.02) |
| Hemisphere | β | −478.00*** | −103.17*** | 0.02 |
| | SE | (46.38) | (15.85) | (0.02) |
| Language experience × posterior STG | β | −54.04 | −20.53 | −0.02 |
| | SE | (90.37) | (30.88) | (0.05) |
| Language experience × anterior STS | β | −104.26 | −28.61 | −0.01 |
| | SE | (90.37) | (30.88) | (0.05) |
| Language experience × middle STS | β | 23.56 | 28.91 | −0.02 |
| | SE | (90.37) | (30.88) | (0.05) |
| Language experience × posterior STS | β | 40.34 | 38.03 | −0.05 |
| | SE | (90.37) | (30.88) | (0.05) |
| Language experience × PT | β | −172.20. | −35.12 | −0.12* |
| | SE | (90.37) | (30.88) | (0.05) |
| Language experience × HG | β | −39.06 | −4.31 | −0.02 |
| | SE | (90.37) | (30.88) | (0.05) |

| | | Volume | Area | Thickness |
|---|---|---|---|---|
| Language experience × HS | β | –82.13 | –25.18 | –0.01 |
| | SE | (90.37) | (30.88) | (0.05) |
| Language experience × PP | β | –91.67 | –9.65 | –0.07 |
| | SE | (90.37) | (30.88) | (0.05) |
| Language experience × hemisphere | β | 41.88 | –17.55 | 0.02 |
| | SE | (90.37) | (30.88) | (0.05) |
| Posterior STG × hemisphere | β | 407.62*** | 99.63*** | 0.02 |
| | SE | (65.60) | (22.42) | (0.03) |
| Anterior STS × hemisphere | β | 371.93*** | 67.74** | –0.05 |
| | SE | (65.60) | (22.42) | (0.03) |
| Middle STS × hemisphere | β | 440.40*** | 43.39. | 0.11** |
| | SE | (65.60) | (22.42) | (0.03) |
| Posterior STS × hemisphere | β | 1186.10*** | 435.79*** | 0.01 |
| | SE | (65.60) | (22.42) | (0.03) |
| PT × hemisphere | β | 222.12*** | 7.78 | 0.04 |
| | SE | (65.60) | (22.42) | (0.03) |
| HG × hemisphere | β | 255.17*** | 19.78 | 0.09** |
| | SE | (65.60) | (22.42) | (0.03) |
| HS × hemisphere | β | 363.43*** | 23.36 | 0.16*** |
| | SE | (65.60) | (22.42) | (0.03) |
| PP × hemisphere | β | 486.42*** | 146.40*** | –0.15*** |
| | SE | (65.60) | (22.42) | (0.03) |
| Language experience × posterior STG × hemisphere | β | –36.81 | 32.36 | –0.02 |
| | SE | (127.81) | (43.68) | (0.07) |
| Language experience × anterior STS × hemisphere | β | 75.05 | 52.32 | 0.01 |
| | SE | (127.81) | (43.68) | (0.07) |
| Language experience × middle STS × hemisphere | β | 15.40 | 16.32 | 0.01 |
| | SE | (127.81) | (43.68) | (0.07) |
| Language experience × posterior STS × hemisphere | β | 50.84 | 52.48 | –0.03 |
| | SE | (127.81) | (43.68) | (0.07) |
| Language experience × PT × hemisphere | β | 31.46 | 33.64 | 0.03 |
| | SE | (127.81) | (43.68) | (0.07) |
| Language experience × HG × hemisphere | β | –71.22 | 8.23 | 0.00 |
| | SE | (127.81) | (43.68) | (0.07) |
| Language experience × HS × hemisphere | β | –13.56 | 24.11 | 0.02 |
| | SE | (127.81) | (43.68) | (0.07) |
| Language experience × PP × hemisphere | β | –37.13 | 13.98 | 0.02 |
| | SE | (127.81) | (43.68) | (0.07) |
| SD (Intercept id) | | 91.84 | 24.92 | 0.06 |
| SD (Observations) | | 382.49 | 130.71 | 0.20 |

| | Volume | Area | Thickness |
|---|---|---|---|
| *Num.Obs.* | *2448* | *2448* | *2448* |
| *R² Marg.* | *0.904* | *0.937* | *0.793* |
| *R² Cond.* | *0.910* | *0.939* | *0.812* |
| *AIC* | *35864.3* | *30661.7* | *−580.1* |
| *BIC* | *36102.2* | *30899.7* | *−342.2* |
| *ICC* | *0.1* | *0.0* | *0.1* |
| *RMSE* | *374.14* | *128.27* | *0.19* |

STG: superior temporal gyrus, STS: superior temporal sulcus, PT: planum temporale, HG: Heschl's gyrus, HS: Heschl's sulcus, PP: planum polare.
p<0.1, +p = 0.05, *p<0.05, **p<0.01, ***p<0.001.

## 5 Superior temporal region and language experience

**Appendix 1—table 3.** Results of the linear mixed models testing the effect of language experience on the structure (volume, area, and average thickness) of the gyri in the superior temporal region: first, second, and third transverse temporal gyrus (TTG).
Anterior TTG was used as the reference level.

| | | Volume | Area | Thickness |
|---|---|---|---|---|
| (Intercept) | β | −86.44*** | −31.09*** | 0.05** |
| | SE | (14.45) | (4.14) | (0.01) |
| Language experience | β | −4.83 | 0.92 | −0.06* |
| | SE | (32.25) | (9.24) | (0.03) |
| Age | β | −1.33 | −0.30 | 0.00 |
| | SE | (0.86) | (0.22) | (0.00) |
| Sex | β | 14.78 | 2.72 | 0.01 |
| | SE | (12.69) | (3.66) | (0.01) |
| Whole-brain: volume/area/thickness | β | 0.00*** | 0.00*** | 0.98*** |
| | SE | (0.00) | (0.00) | (0.16) |
| Second gyrus | β | 351.19*** | 119.93*** | −0.13*** |
| | SE | (17.05) | (4.89) | (0.01) |
| Third gyrus | β | −169.80*** | −50.83*** | 0.01 |
| | SE | (17.55) | (5.03) | (0.02) |
| Hemisphere | β | 51.04*** | 19.58*** | −0.04** |
| | SE | (14.42) | (4.13) | (0.01) |
| Language experience × second gyrus | β | −30.18 | −14.89 | 0.09** |
| | SE | (36.03) | (10.33) | (0.03) |
| Language experience × third gyrus | β | −5.66 | 3.50 | −0.05 |
| | SE | (36.88) | (10.57) | (0.03) |
| Language experience × hemisphere | β | −27.16 | −7.25 | 0.00 |
| | SE | (31.46) | (9.02) | (0.03) |
| Second gyrus × hemisphere | β | 8.34 | 8.32. | −0.02 |
| | SE | (17.01) | (4.88) | (0.01) |

*Appendix 1—table 3 Continued on next page*

*Appendix 1—table 3 Continued*

|  |  | Volume | Area | Thickness |
|---|---|---|---|---|
| Third gyrus × hemisphere | β | –18.79 | –6.34 | –0.01 |
|  | SE | (17.55) | (5.03) | (0.02) |
| Language experience × second gyrus × hemisphere | β | –11.32 | –2.86 | –0.02 |
|  | SE | (36.04) | (10.34) | (0.03) |
| Language experience × third gyrus × hemisphere | β | 30.25 | 11.01 | –0.03 |
|  | SE | (36.88) | (10.57) | (0.03) |
| SD (Intercept id) |  | 0.01 | 0.00 | 0.07 |
| SD (Observations) |  | 257.90 | 73.94 | 0.22 |
| Num.Obs. |  | 567 | 567 | 567 |
| $R^2$ Marg. |  | 0.521 | 0.596 | 0.254 |
| $R^2$ Cond. |  | 0.521 |  | 0.324 |
| AIC |  | 7836.9 | 6455.5 | 66.9 |
| BIC |  | 7910.7 | 6529.3 | 140.7 |
| ICC |  | 0.0 |  | 0.1 |
| RMSE |  | 254.47 | 72.95 | 0.21 |

p<0.1, +p = 0.05, *p<0.05, **p<0.01, ***p<0.001.

## 6 Vertex-wise analysis

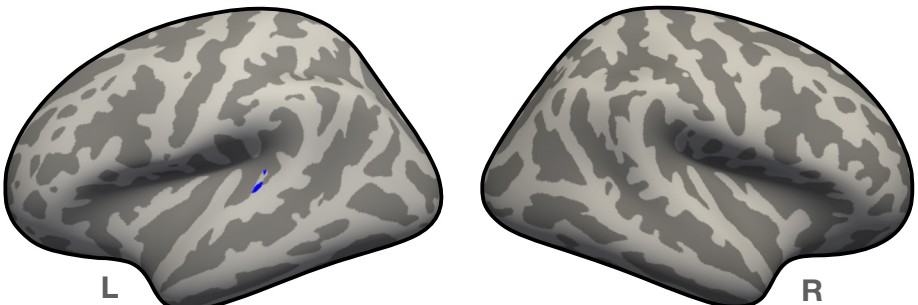

**Appendix 1—figure 5.** Results of a whole-brain vertex-wise analysis, aimed at establishing relations between the language experience index and whole-brain cortical thickness. Overlaid on the inflated surface of the fsaverage template brain is the thresholded at p<0.0001 (uncorrected) significance map from the conducted *F*-test showing a negative relationship between cortical thickness in the highlighted region and the degree of multilingual language experience.

## 7 Second TTG and effects of language typology

Kolmogorov-Smirnov normality tests were run on all models' residuals, revealing that parametric linear models were appropriate for the present data (all ps>0.4). According to a frequentist analysis of the data, all eight overall regression models were statistically significant: (1) Left hemisphere: $F(4,125)$ = 7.802, p<0.001, $F(4,125)$ = 6.32, p<0.001, $F(4,125)$ = 8.287, p<0.001, $F(4,125)$ = 6.98, p<0.001, respectively for the models with the four language experience indices: (a) the cumulative language experience measure not accounting for typology, and cumulative language experience weighted by overlaps between languages at the level of (b) acoustic/articulatory features, (c) phonemes, and (d) counts of phonological classes; and (2) Right hemisphere: $F(4,91)$ = 4.295, p=0.003, $F(4,91)$ = 3.461, p=0.01, $F(4,91)$ = 4.331, p=0.002, $F(4,91)$ = 3.722, p=0.007, respectively, for the models with the four language experience indices: (a) the cumulative language experience measure not accounting for typology, and cumulative language experience weighted by overlaps between languages at the level of (b) acoustic/articulatory features, (c) phonemes, and (d) counts of phonological classes.

## 7.1 Cumulative phoneme inventory and the second TTG

The two models including the 'cumulative phoneme inventory' measure were statistically significant: (1) Left hemisphere: $F(5,124) = 8.308$, $p<0.001$, and (2) Right hemisphere: $F(5,90) = 3.42$, $p=0.007$, see further *Appendix 1—table 4*. *Appendix 1—figure 6* presents the relationship between the thickness of the second TTG (left and right) and the cumulative phoneme inventory.

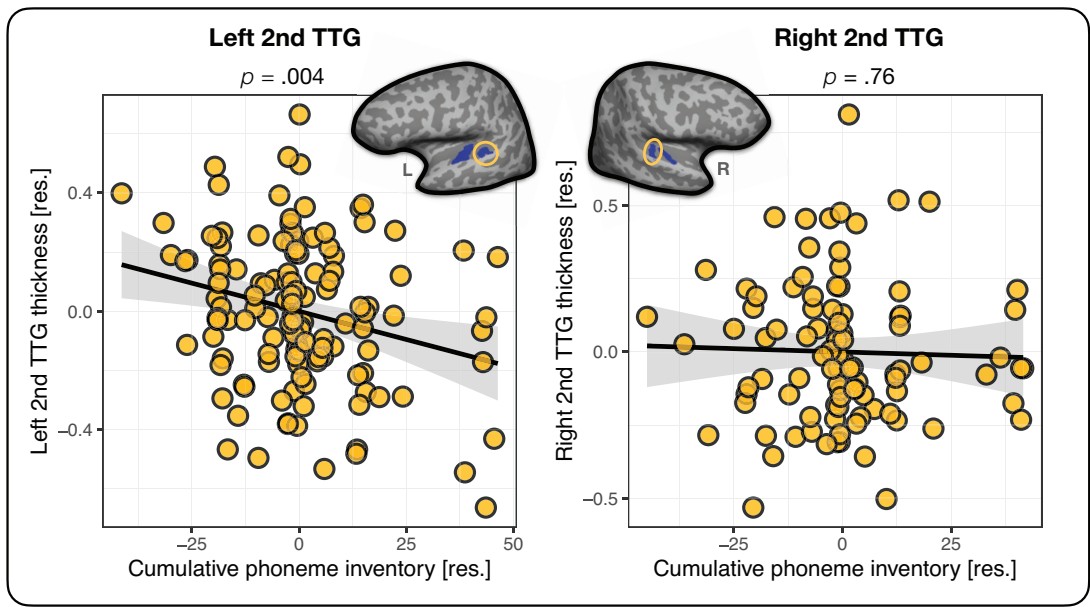

**Appendix 1—figure 6.** Cumulative phoneme inventory and thickness of the second transverse temporal gyrus (TTG). Average thickness of the second TTG in the left and right hemisphere in relation to the number of unique phonemes each participant was exposed to across all their languages (the plotted values are residuals controlled for age, sex, mean hemispheric thickness, and the language experience index irrespective of typology).

**Appendix 1—table 4.** Left and right second transverse temporal gyri (TTGs) and cumulative phoneme inventory.

Multiple regression model parameters (parameter estimates and standard errors, in brackets; p-values are listed according to the coding presented underneath the table) for the average cortical thickness of the second TTG (left and right), as predicted by the 'cumulative phoneme inventory' index. For comparison, models with the cumulative language experience measure not accounting for typology, and cumulative language experience weighted by overlaps between languages at the level of phonemes are also reported. Last two rows present model comparison results (additional variance explained and $BF_{10}$ and $BF_{01}$ values are also reported).

| | | Left | | | Right | | |
|---|---|---|---|---|---|---|---|
| Language experience Models: | | Cumulative phoneme inventory | Language experience | Language experience (phonemes) | Cumulative phoneme inventory | Language experience | Language experience (Phonemes) |
| (Intercept) | β | 0.00 | 0.00 | 0.00 | 0.00 | 0.00 | 0.00 |
| | SE | (0.02) | (0.02) | (0.02) | (0.03) | (0.03) | (0.03) |
| Language experience | β | 0.05 | –0.12** | – | –0.08 | –0.10+ | – |
| | SE | (0.07) | (0.05) | – | (0.09) | (0.05) | – |
| Phonemes | β | 0.00** | – | – | 0.00 | – | – |
| | SE | (0.00) | – | – | (0.00) | – | – |

*Appendix 1—table 4 Continued on next page*

*Appendix 1—table 4 Continued*

| | | Left | | | Right | | |
|---|---|---|---|---|---|---|---|
| Language experience (phonemes) | β | – | – | –0.35** | - | - | –0.26+ |
| | SE | – | - | (0.12) | - | - | (0.13) |
| Age | β | 0.00. | 0.00 | 0.00 | 0.00 | 0.00 | 0.00 |
| | SE | (0.00) | (0.00) | (0.00) | (0.00) | (0.00) | (0.00) |
| Mean thickness (left/right) | β | 1.43*** | 1.40*** | 1.37*** | 0.95* | 0.94* | 0.92* |
| | SE | (0.29) | (0.30) | (0.30) | (0.37) | (0.37) | (0.37) |
| Sex | β | 0.00 | 0.00 | 0.00 | 0.01 | 0.00 | 0.01 |
| | SE | (0.02) | (0.02) | (0.02) | (0.03) | (0.03) | (0.03) |
| Num.Obs. | | 130 | 130 | 130 | 96 | 96 | 96 |
| $R^2$ | | 0.251 | 0.200 | 0.210 | 0.160 | 0.159 | 0.160 |
| $R^2$ Adj. | | 0.221 | 0.174 | 0.184 | 0.113 | 0.122 | 0.123 |
| AIC | | 11.9 | 18.5 | 16.9 | 13.8 | 11.9 | 11.8 |
| BIC | | 31.9 | 35.7 | 34.1 | 31.8 | 27.3 | 27.2 |
| Log.Lik. | | 1.062 | –3.232 | –2.430 | 0.085 | 0.036 | 0.101 |
| F | | 8.308 | 7.802 | 8.287 | 3.420 | 4.295 | 4.331 |
| RMSE | | 0.24 | 0.25 | 0.25 | 0.24 | 0.24 | 0.24 |
| $\Delta R^2$ Adjusted | | – | –0.04 | –0.03 | – | 0.01 | 0.01 |
| $BF_{10}$ | | – | 0.15 | 0.34 | – | 9.33 | 9.96 |
| $BF_{01}$ | | – | 6.43 | 2.88 | – | 0.11 | 0.10 |

p<0.1, +p = 0.05, *p<0.05, **p<0.01, ***p<0.001.

## 8 Language experience in participants with a single TTG

Kolmogorov-Smirnov normality tests were run on all models' residuals, revealing that parametric linear models were appropriate for the present data (all ps>0.29). According to a frequentist analysis of the data, all eight overall regression models were statistically significant: (1) Right HG: $F_{(4,35)}$ = 6.365, p<0.001, $F_{(4,35)}$ = 4.851, p=0.003, $F_{(4,35)}$ = 5.403, p=0.002, $F_{(4,35)}$ = 5.72, p=0.001, respectively for models with the four language experience indices: (a) the cumulative language experience measure not accounting for typology, and cumulative language experience weighted by overlaps between languages at the level of (b) acoustic/articulatory features, (c) phonemes, and (d) counts of phonological classes; and (2) Right PT: $F_{(4,35)}$ = 2.655, p=0.049, $F_{(4,35)}$ = 2.639, p=0.050, $F_{(4,35)}$ = 2.66, p=0.048, $F_{(4,35)}$ = 2.685, p=0.049, respectively for models with the four language experience indices: (a) the cumulative language experience measure not accounting for typology, and cumulative language experience weighted by overlaps between languages at the level of (b) acoustic/articulatory features, (c) phonemes, and (d) counts of phonological classes.

## 9 Effects of language proficiency

The findings of thinner cortex in the second TTG (when present) were further followed up by an analysis investigating the effect of proficiency attained by the participants in their individual languages. Here, we again constructed cumulative language experience measures per participant using the Shannon's entropy equation *Shannon's, 1948*, but this time from the self-reported proficiency ratings (on a scale from 0 to 10) of each of participants' languages. First, we computed a general proficiency score and used it in two linear models to establish whether the AoA-derived cumulative language experience measure could be replicated by the proficiency-derived measure, since AoA and proficiency are known to be highly correlated. This analysis therefore served as a

**Appendix 1—table 5.** Right superior temporal plane (Heschl's gyrus and planum temporale) and language experience in participants with one transverse temporal gyrus (TTG).

Multiple regression model parameters (parameter estimates and standard errors, in brackets; p-values are listed according to the coding presented underneath the table) for the average cortical thickness of the right Heschl's gyrus, and the right planum temporale, as predicted by the four language experience indices: (1) the cumulative language experience measure not accounting for typology, and cumulative language experience weighted by overlaps between languages at the level of (2) phonemes, (3) acoustic/articulatory features, and (4) counts of phonological classes. Last two rows present model comparison results (additional variance explained and $BF_{10}$ values). NB. All models including typological information were compared against the 'No typology' model.

| | | Right Heschl's gyrus | | | | Right planum temporale | | | |
|---|---|---|---|---|---|---|---|---|---|
| | | No typology | Features | Phonemes | Phonological classes | No typology | Features | Phonemes | Phonological classes |
| (Intercept) | β | −0.01 | −0.01 | −0.01 | −0.13* | 2.52*** | 2.52*** | 2.52*** | 2.50*** |
| | SE | (0.03) | (0.03) | (0.03) | (0.06) | (0.03) | (0.03) | (0.03) | (0.07) |
| Language experience indices | β | 0.15** | 0.48+ | 0.35* | 0.82* | 0.02 | 0.03 | 0.06 | 0.11 |
| | SE | (0.05) | (0.24) | (0.15) | (0.32) | (0.07) | (0.30) | (0.19) | (0.41) |
| Age | β | 0.00 | 0.00 | 0.00 | 0.00 | −0.00 | −0.00 | −0.00 | −0.00 |
| | SE | (0.00) | (0.00) | (0.00) | (0.00) | (0.00) | (0.00) | (0.00) | (0.00) |
| Sex | β | 0.05 | 0.03 | 0.05 | 0.04 | 0.06 | 0.05 | 0.06 | 0.06 |
| | SE | (0.05) | (0.06) | (0.06) | (0.05) | (0.07) | (0.07) | (0.07) | (0.07) |
| Mean Thickness (right) | β | 0.93** | 0.99** | 0.96** | 0.93** | 0.90* | 0.90* | 0.91* | 0.90* |
| | SE | (0.30) | (0.31) | (0.30) | (0.30) | (0.38) | (0.39) | (0.38) | (0.38) |
| Num.Obs. | | 40 | 40 | 40 | 40 | 40 | 40 | 40 | 40 |
| R² | | 0.42 | 0.36 | 0.38 | 0.40 | 0.23 | 0.23 | 0.23 | 0.23 |
| R² Adj. | | 0.36 | 0.28 | 0.31 | 0.33 | 0.15 | 0.14 | 0.15 | 0.15 |
| AIC | | −36.6 | −32.4 | −34.0 | −34.9 | −15.6 | −15.5 | −15.6 | −15.6 |
| BIC | | −26.5 | −22.3 | −23.8 | −24.7 | −5.4 | −5.4 | −5.5 | −5.4 |
| Log.Lik. | | 24.30 | 22.19 | 22.99 | 23.43 | 13.79 | 13.76 | 13.81 | 13.79 |
| RMSE | | 0.13 | 0.14 | 0.14 | 0.13 | 0.17 | 0.17 | 0.17 | 0.17 |

*Appendix 1—table 5 continued on next page*

*Appendix 1—table 5 continued*

| | **Right Heschl's gyrus** | | | **Right planum temporale** | | |
|---|---|---|---|---|---|---|
| $\Delta R^2$ Adjusted | – | –0.07 | –0.04 | –0.03 | – | >–0.01 | <0.01 | <0.01 |
| $BF_{10}$ | – | 0.12 | 0.27 | 0.42 | – | 0.97 | 1.02 | 1.01 |

p<0.1, +p=0.05, *p<0.05, **p<0.01, ***p<0.001.

'sanity check' for the results obtained from the analyses reported in section 'Superior temporal plane and language experience' of the main manuscript. Indeed, we observed that the thickness of the second TTG (when present) was negatively related to the cumulative language experience measure derived from proficiency ratings. This effect was significant for the left hemisphere second TTG thickness, and showed a trend toward significance for the right ($\beta = -0.11$, $t = -2.41$, p=0.018, small effect size: $f^2 = 0.05$; and $\beta = -0.10$, $t = -1.87$, p=0.065, small effect size: $f^2 = 0.05$, for left and right second TTG, respectively).

Given the hypothesis that multilingualism is a dynamic process reshaping brain structure and inducing both increases (in initial stages) and decreases (in peak efficiency) in brain morphological indices (*Pliatsikas, 2020*), we further explored the effect of proficiency of different languages by investigating the role of languages spoken proficiently and non-proficiently on the structure of the second TTG. Here, we computed two further different indices per participant: (1) derived only from languages spoken proficiently (rated at 7 and higher out of 10 on a self-reported proficiency scale), and (2) derived from languages spoken at a low level of proficiency (rated lower than 7 out of 10). We subsequently used these indices in two further analyses to test the hypothesis that decreases in brain morphological indices (here in cortical thickness) are driven by peak efficiency of language functioning (operationalized here as high proficiency in many languages). We did observe that the thickness of the second TTG was related to the measure of language experience derived from languages spoken proficiently, however this effect was only significant in the right hemisphere ($\beta = -0.18$, $t = -2.89$, p=0.005, small effect size: $f^2 = 0.11$; and $\beta = -0.02$, $t = -0.44$, p=0.66, for right and left second TTG, respectively). Second TTG's thickness was not related to the language experience measure derived from languages spoken at low level of proficiency ($\beta = -0.01$, $t = -0.11$, p=0.91; and $\beta = -0.12$, $t = -1.69$, p=0.097 for right and left second TTG, respectively), see *Appendix 1—figure 7*.

## PROFICIENCY-BASED LANGUAGE BACKGROUND
## IN PARTICIPANTS WITH MULTIPLE TTGs

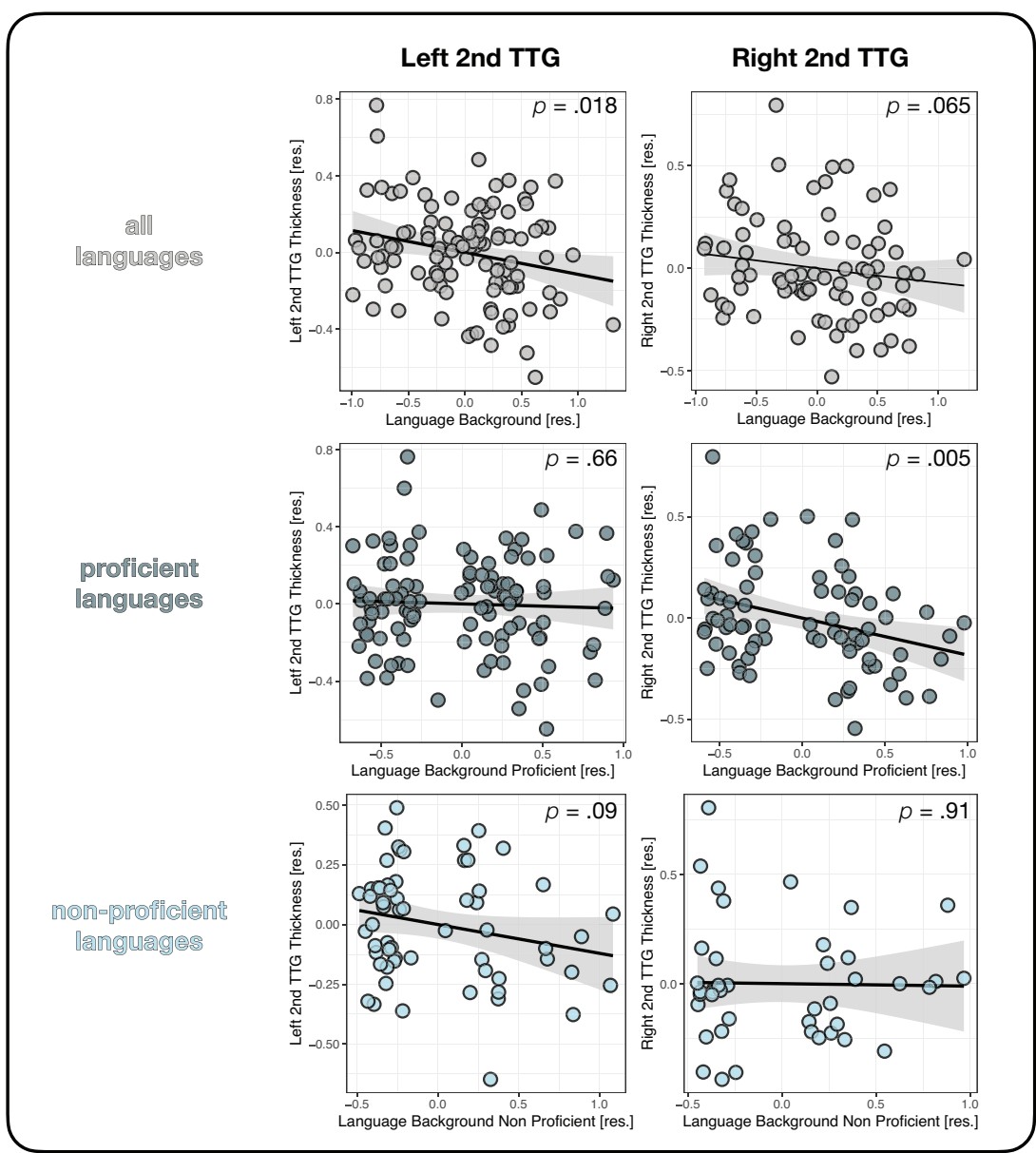

**Appendix 1—figure 7.** Multilingual proficiency and thickness of the second transverse temporal gyrus (TTG). Average thickness of the second TTG in the left and right hemisphere as a function of language proficiency in all languages of each participant (top panel), only their proficient languages (middle panel), and only their non-proficient languages (bottom panel). Plots show residuals, controlling for age, sex, and mean hemispheric thickness.

## 10 Replication analysis

Kolmogorov-Smirnov normality tests were run on all models' residuals, revealing that parametric linear models were appropriate for the present data (all ps>0.5). According to a frequentist analysis of the data, all four overall regression models for left hemisphere data were statistically not significant: $F(5,55) = 1.109$, p<0.001, $F(5,55) = 1.12$, p=0.36, $F(5,55) = 8.287$, p=0.37, $F(5,55) = 1.072$, p=0.38, respectively, for models with the four language experience indices: (a) the cumulative language experience measure not accounting for typology, and cumulative language experience weighted by overlaps between languages at the level of (b) acoustic/articulatory features, (c) phonemes, and (d) counts of phonological classes. Models for the right hemisphere data were all statistically significant,

apart from the counts of phonological classes model: $F(5,48) = 2.781$, p=0.03, $F(5,48) = 5.158$, p<0.001, $F(5,48) = 3.244$, p=0.013, $F(5,48) = 2.342$, p=0.06, respectively, for models with the four language experience indices: (a) the cumulative language experience measure not accounting for typology, and cumulative language experience weighted by overlaps between languages at the level of (b) acoustic/articulatory features, (c) phonemes, and (d) counts of phonological classes.

## 11 Phonological classes as defined by *Dediu and Moisik, 2016*

(v = vowels, c = consonants) segments, vowels, monophtongs, diphtongs, triphtongs, heights.v, lengths.v, long.v, nasal.v, round.v, high.v, mid.v, low.v, front.v, back.v, tense.v, lax.v, atr.v, rtr.v, raised.v, retracted.v, fronted.v, glottalized.v, unique.v, unique.nasal.v, heights_mono.v, heights_di.v, heights_tri.v, lengths_mono.v, lengths_di.v, lengths_tri.v, long_mono.v, long_di.v, long_tri.v, nasal_mono.v, nasal_di.v, nasal_tri.v, round_mono.v, round_di.v, round_tri.v, high_mono.v, high_di.v, high_tri.v, mid_mono.v, mid_di.v, mid_tri.v, low_mono.v, low_di.v, low_tri.v, front_mono.v, front_di.v, front_tri.v, back_mono.v, back_di.v, back_tri.v, tense_mono.v, tense_di.v, tense_tri.v, lax_mono.v, lax_di.v, lax_tri.v, atr_mono.v, atr_di.v, atr_tri.v, rtr_mono.v, rtr_di.v, rtr_tri.v, raised_mono.v, raised_di.v, raised_tri.v, retracted_mono.v, retracted_di.v, retracted_tri.v, fronted_mono.v, fronted_di.v, fronted_tri.v, glottalized_mono.v, glottalized_di.v, glottalized_tri.v, unique_mono.v, unique_di.v, unique_tri.v, unique.nasal_mono.v, unique.nasal_di.v, unique.nasal_tri.v, consonants, places.c, bilabial.c, labiodental.c, dental.c, alveolar.c, dental_alveolar.c, palatoalveolar.c, alveolopalatal.c, postalveolar.c, true_retroflex.c, palatal.c, velar.c, uvular.c, pharyngeal_epiglottal.c, glottal.c, labial.c, coronal.c, dorsal.c, guttural.c, manners.c, obstruent.c, voiced_obstruent.c, voiceless_obstruent.c, aspirated_obstruent.c, glottalized_obstruent.c, stop.c, voiced_stop.c, voiceless_stop.c, aspirated_stop.c, glottalized_stop.c, fricative.c, voiced_fricative.c, voiceless_fricative.c, affricate.c, sonorant.c, voiced_sonorant.c, voiceless_sonorant.c, glottalized_resonant.c, nasal.c, approximant.c, tapflap.c, trill.c, trill_tap.c, coronal_trill_tap.c, glottalized.c, uvt.c, uvt.stops.c, uvt.fricatives.c, uvt.affricates.c, uvt.nasals.c, uvt.approximants.c, lvt.c, lvt.stops.c, lvt.fricatives.c, lvt.affricates.c, lvt.nasals.c, lvt.approximants.c, ratio.voiced.voiceless.obstruents, ratio.voiced.voiceless.stops, ratio.obstruents.sonorants, egressive, implosive, ejective, click, voiceless, voiced, breathy, creaky, tones, bilabial.fricatives, labiodental.fricatives, alveolar.fricatives, nonsibilant.dental.fricatives, sibilant.dental.fricatives, bilabiallabiodental.affricates, bilabial.affricates, retroflex.stops, retroflex.fricatives, retroflex.affricates, retroflex.nasals, retroflex.approximants

