## [Editor Report · eLife Assessment]

This report details **convincing** evidence that experience with multilingualism in general, and with larger phonological inventories specifically, is related to differences in the structure of the transverse temporal gyri. The project is notable for using a relatively large sample, and confirming the primary finding in a second sample. The **important** findings strongly point to experience-dependent plasticity related to language experience as a driver of neuroanatomy of the auditory cortex.

---

## [Referee Report · Reviewer #1 (Public review)]

Summary:

The goal of this project is to test the hypothesis that individual differences in experience with multiple languages relate to differences in brain structure, specifically in the transverse temporal gyrus. The approach used here is to focus specifically on the phonological inventories of these languages, looking at the overall size of the phonological inventory as well as the acoustic and articulatory diversity of the cumulative phonological inventory in people who speak one or more languages. The authors find that the thickness of the transverse temporal gyrus (either the primary TTG, in those with one TTG, or in the second TTG, in people with multiple gyri) was related to language experience, and that accounting for the phonological diversity of those languages improved the model fit. Taken together, the evidence suggests that learning more phonemes (which is more likely if one speaks more than one language) leads to experience-related plasticity brain regions implicated in early auditory processing.

Strengths:

This project is rigorous in its approach--not only using a large sample but replicating the primary finding in a smaller, independent sample. Language diversity is difficult to quantify, and likely to be qualitatively and quantitatively distinct across different populations, and the authors use a custom measure of multilingualism (accounting for both number of languages as well as age of acquisition) and three measures of phonological diversity. The team has been careful in discussion of these findings, and while it is possible that pre-existing differences in brain structure could lead to an aptitude difference which could drive one to learn more than one language, the fine-grained relationships with phonological diversity seem less likely to emerge from aptitude rather than experience.

The authors have satisfied my curiosity regarding other potential confounds in the data, including measurements of lexical distance as well as phonological typology.

---

## [Referee Report · Reviewer #2 (Public review)]

This work investigates the possible association between language experience and morphology of the superior temporal cortex, a part of the brain responsible for the processing of auditory stimuli. Previous studies have found associations between language and music proficiency as well as language learning aptitude and cortical morphometric measures in regions in the primary and associated auditory cortex. These studies have most often, however, focused on finding neuroanatomical effects of difference between features in a few (often two) languages or from learning single phonetic/phonological features and have often been limited in terms of N. On this background, the authors use more sophisticated measures of language experience that take into account the age of onset and the differences in phonology between languages the subjects have been exposed as well as a larger number of subjects (N = 146 + 69) to relate language experience to the shape and structure of the superior temporal cortex, measured from T1-weighted MRI data. It shows solid evidence for there being a negative relationship between language experience and the right 2nd transverse temporal gyrus as well as some evidence for the relationship representing phoneme-level cross-linguistic information.

Strengths

The use of entropy measures to quantify language experience and include typological distance measures allows for a more general interpretation of the results and is an important step toward respecting and making use of linguistic diversity in neurolinguistic experiments.

A relatively large group of subjects with a range of linguistic backgrounds.

The full analysis of the structure of the superior temporal cortex including cortical volume, area, as well as the shape of the transverse gyrus/gyri. There is a growing literature on the meaning of the shape and number of the transverse gyri in relation to language proficiency and the authors explore all measures given the available data.

The authors chose to use a replication data set to verify their data, which is applaudable. However, see the relevant point under "Weaknesses".

Weaknesses

Even if the language experience and typological distance measures are a step in the right direction for correctly associating language exposure with cortical plasticity, it still is a measure that is insensitive to the intensity of the exposure.

Only the result from the multiple transverse temporal gyri (2nd TTG) is analyzed in the replicated dataset. Only the association in the right hemisphere 2nd TTG is replicated but this is not reflected in the discussion or the conclusions. The positive correlation in the right TTG is thus not attempted to be replicated.

The replication dataset differed in more ways than the more frequent combination of English and German experience, as mentioned in the discussion. Specifically, the fraction of monolinguals was higher in the replication dataset and the samples came from different scanners. It would be better if the primary and replication datasets were more equally matched.

---

## [Referee Report · Reviewer #3 (Public review)]

Summary:

The study uses structural MRI to identify how the number, degree of experience, and phonemic diversity of language(s) that a speaker knows can influence the thickness of different sub-segments of auditory cortex. In both a primary and replication sample of adult speakers, the authors find key differences in cortical thickness within specific subregions of cortex due to either the age at which languages are acquired (degree of experience) or the diversity of the phoneme inventories carried by that/those language(s) (breadth of experience).

Strengths:

The results are first and foremost quite fascinating and I do think they make a compelling case for the different ways in which linguistic experience shapes auditory cortex.

The study uses a number of different measures to quantify linguistic experience, related to how many languages a person knows (taking into account the age at which each was learned) as well as the diversity of the phoneme inventories contained within those languages. The primary sample is moderately large for a study that focuses on brain-behaviour relationships; a somewhat smaller replication sample is also deployed in order to test the generality of the effects.

Analytic approaches benefit from the careful use of brain segmentation techniques that nicely capture key landmarks and account for vagaries in the structure of STG that can vary across individuals (e.g., the number of transverse temporal gyri varies from 1-4 across individuals).

Weaknesses:

The specificity of these effects is interesting; some effects really do appear to be localized to left hemisphere and specific subregions of auditory cortex e.g., TTG. There is an ancillary analysis that examines regions outside auditory cortex to examine whether these are the only brain regions for which such effects occur. Expanding the search space to a whole-brain analysis, and a more lenient statistical threshold, does reveal only small patches of the brain outside auditory cortex show similar effects. Notably, these could be due to inflated type-1 error, but overall we would need a much larger sample to be certain.

Discussion of potential genetic differences underlying the findings is interesting. It does represent one alternative account that does not have to do with plasticity/experience, as the authors acknowledge.

The replication sample is useful and a great idea. It does however feature roughly half the number of participants. As the authors are careful to point out, that statistical power is weaker and given small effects in some cases we should not be surprised that the results only partially replicated in that sample.

---

## [Author Response]

The following is the authors’ response to the original reviews.

**Reviewer #1 (Public Review):**
Summary:The goal of this project is to test the hypothesis that individual differences in experience with multiple languages relate to differences in brain structure, specifically in the transverse temporal gyrus. The approach used here is to focus specifically on the phonological inventories of these languages, looking at the overall size of the phonological inventory as well as the acoustic and articulatory diversity of the cumulative phonological inventory in people who speak one or more languages. The authors find that the thickness of the transverse temporal gyrus (either the primary TTG, in those with one TTG, or in the second TTG, in people with multiple gyri) was related to language experience, and that accounting for the phonological diversity of those languages improved the model fit. Taken together, the evidence suggests that learning more phonemes (which is more likely if one speaks more than one language) leads to experience-related plasticity in brain regions implicated in early auditory processing.Strengths:This project is rigorous in its approach--not only using a large sample, but replicating the primary finding in a smaller, independent sample. Language diversity is difficult to quantify, and likely to be qualitatively and quantitatively distinct across different populations, and the authors use a custom measure of multilingualism (accounting for both number of languages as well as age of acquisition) and three measures of phonological diversity. The team has been careful in discussion of these findings, and while it is possible that pre-existing differences in brain structure could lead to an aptitude difference which could drive one to learn more than one language, the fine-grained relationships with phonological diversity seem less likely to emerge from aptitude rather than experience.Weaknesses:It is a bit unclear how the measures of phonological diversity relate to one another--they are partially separable, but rest on the same underlying data (the phonemes in each language). It would be helpful for the reader to understand how these measures are distributed (perhaps in a new figure), and the degree to which they are correlated with one another.

Thank you for the comment. Indeed our description missed this important detail that we now included in the manuscript. Unsurprisingly, the distances all correlated with one another, which we present in Table 2 in Section 2.3 of the revised manuscript. We have also added a figure with distributions of the three distance measures (see Figure S3).

Further, as the authors acknowledge, it is always possible that an unseen factor instead drives these findings--if typological lexical distance measures are available, it would be helpful to enter these into the model to confirm that phonological factors are the specific driver of TTG differences and not language diversity in a more general sense. That said, the relationship between phonological diversity and TTG structure is intuitive.

Thank you for the suggestion. To further establish that our results reflected the relationship between TTG structure and phonological diversity specifically (as opposed to language diversity in a more general sense), we derived a fourth measure of language experience, where the AoA index of different languages was weighted by lexical distances between the languages. Here, we followed the methodology described in Kepinska, Caballero, et al. (2023): We used Levenshtein Distance Normalized Divided (LDND) Wichmann et al., 2010 which was computed using the ASJP.R program by Wichmann (https://github.com/Sokiwi/InteractiveASJP01). Information on lexical distances was combined with language experience information per participant using Rao's quadratic entropy equation in the same way as for the phonological measures.

We then entered this language experience measure accounting for lexical distances between the languages into linear models predicting the thickness of the second left and right TTG (controlling for participants’ age, sex and mean hemispheric thickness) in the main sample, and compared these models with the corresponding models including the original three phonological distance measures (models 24 in Author response table 1), and the measure with no typological information (1).

Below, we list adjusted R2 values of all models, from which it is clear that the index of multilingual language experience accounting for lexical distances between languages (5) explained less variance than the index incorporating phoneme-level distances between languages (2), both in the left and the right hemisphere. This further strengthens our conclusion that our results reflected the relationship between TTG structure and phonological diversity specifically, as opposed to language diversity in a more general sense.

**Author response table 1. sa4table1:** 

	Model	Adjusted R^2^
left 2ndTTG	(1) No typology	0.174
(2) Phonemes	0.184
(3) Features	0.142
(4) Phonological classes	0.156
(5) Lexical	0.164
right 2ndTTG	(1) No typology	0.122
(2) Phonemes	0.123
(3) Features	0.094
(4) Phonological classes	0.103
(5) Lexical	0.116

We have added a description of this analysis to the manuscript, Section 3.3, lines 357-370.

One curious aspect of this paper relates to the much higher prevalence of split or duplicate TTG in the sample. The authors do a good job speculating on how features of the TASH package might lead to this, but it is unclear where the ground truth lies--some discussion of validation of TASH against a gold standard would be useful.

The validation of the TASH toolbox in comparison to gold standard manual measurement involved assessing how well the measurements of left and right Heschl's gyrus (HG) volumes obtained using the TASH method correlated with those obtained through manual labeling (see Dalboni da Rocha et al., 2020 for details). This validation process was conducted across three independent datasets. Additionally, for comparison, the manually labeled HG volumes were also compared with those obtained using FreeSurfer's Destrieux parcellation of the transverse temporal gyrus in the same datasets. The validation process, therefore, involved rigorous comparisons of HG volumes obtained through manual labeling, FreeSurfer, and TASH across different datasets, along with an assessment of inter-rater reliability for the manual labeling procedure. This comprehensive approach ensures that the results are robust and reliable. TASH_complete, the version used in the present work, is an extension of the extensively validated TASH, which apart from the first gyrus, also identifies additional transverse temporal gyri (i.e. Heschl’s gyrus duplications and multiplications) situated in the PT, when present. Since work on the correspondence between manually identified TTG multiplications is still ongoing, as outlined in the Methods section, we complemented the automatic segmentation by extensive visual assessment of the identified posterior gyri. This process involved removing from the analysis those gyri that lay along the portion of the superior temporal plane that curved vertically (i.e., within the parietal extension, Honeycutt et al., 2000), when present. Given that TASH_complete and TASH operate on the same principles and are both based on FreeSurfer’s surface reconstruction and cortical parcellation (which have been extensively validated against manual tracing and other imaging modalities, showing good accuracy), and since we have visually inspected all segmentations, we are confident as to the accuracy of the reported TTG variability. It has to be further noted that the prevalence of TTG multiplications beyond 2nd full posterior duplications was not systematically assessed in previous descriptive reports (Marie, 2015). However, we acknowledge that more work is needed to further ascertain anatomical accuracy of the segmentations, and we elaborate on this point in the Discussion of the revised manuscript (lines 621-623).

**Reviewer #2 (Public Review):**
This work investigates the possible association between language experience and morphology of the superior temporal cortex, a part of the brain responsible for the processing of auditory stimuli. Previous studies have found associations between language and music proficiency as well as language learning aptitude and cortical morphometric measures in regions in the primary and associated auditory cortex. These studies have most often, however, focused on finding neuroanatomical effects of difference between features in a few (often two) languages or from learning single phonetic/phonological features and have often been limited in terms of N. On this background, the authors use more sophisticated measures of language experience that take into account the age of onset and the differences in phonology between languages the subjects have been exposed to as well as a larger number of subjects (N = 146 + 69) to relate language experience to the shape and structure of the superior temporal cortex, measured from T1weighted MRI data. It shows solid evidence for there being a negative relationship between language experience and the right 2nd transverse temporal gyrus as well as some evidence for the relationship representing phoneme-level cross-linguistic information.StrengthsThe use of entropy measures to quantify language experience and include typological distance measures allows for a more general interpretation of the results and is an important step toward respecting and making use of linguistic diversity in neurolinguistic experiments.A relatively large group of subjects with a range of linguistic backgrounds.The full analysis of the structure of the superior temporal cortex including cortical volume, area, as well as the shape of the transverse gyrus/gyri. There is a growing literature on the meaning of the shape and number of the transverse gyri in relation to language proficiency and the authors explore all measures given the available data.The authors chose to use a replication data set to verify their data, which is applaudable. However, see the relevant point under "Weaknesses".WeaknessesThe authors fail to explain how a thinner cortex could reflect the specialization of the auditory cortex in the processing of diverse speech input. The Dynamic Restructuring Model (Pliatsikas, 2020) which is referred to does not offer clear guidance to interpretation. A more detailed discussion of how a phonologically diverse environment could lead to a thinner cortex would be very helpful.

Thank you for bringing our attention to this point. We have now extended the explanation we had previously included in the Discussion by including the following passage on p. 20 (lines 557-566) of the revised manuscript:

“Experience-induced pruning is essential for maintaining an efficient and adaptive neural network. It reinforces relevant neural circuits for faster more efficient information processing, while diminishing those that are less active, or less beneficial. The cortical specialization may need to arise because phonologically more diverse language experience requires that the mapping of acoustic signal to sound categories is denser, more detailed and more intricate. As a result, the brain may need to engage in more intensive processing to discriminate between and accurately perceive the sound categories of each language. This increased cognitive demand may, in turn, require the auditory and language processing regions of the brain to adapt and become more efficient. Over time, this heightened effort for successful speech perception and sound discrimination may lead to neural plasticity, resulting in cortical specialization. This means that cortical areas become more finely tuned and specialized for processing the unique phonological features of language(s) spoken by individuals.”

We have also added a passage to the Introduction regarding the possible microscopic or physiological underpinnings of the brain structural differences that we observe macroscopically using structural MRI (lines 68-73):

“Such environmental effect on cortical thickness might in turn be tied to microstructural changes to the underlying brain tissue, such as modifications in dendritic length and branching, synaptogenesis or synaptic pruning, growth of capillaries and glia, all previously tied to some kind of environmental enrichment and/or skill learning (see Lövdén et al., 2013; Zatorre et al., 2012 for overviews). Increased cortical thickness may reflect synaptogenesis and dendritic growth, while cortical thinning observed with MRI may be a result of increased myelination (Natu et al., 2019) or synaptic pruning.”

It is difficult to understand what measure of language experience is used when. Clearer and more explicit nomenclature would assist in the interpretation of the results.

We have added more explicit list of indices used in the Introduction (lines 104-107 of the revised manuscript) and in Section 2.4 and used them consistently throughout the text:

(1) language experience index not accounting for typological features: ‘Language experience - no typology’

(2) measures combining language experience with typological distances at different levels:

a. ‘Language experience – features’,

b. ‘Language experience – phonemes’,

c. ‘Language experience – phonological classes’.

There is a lack of description of the language backgrounds of the included subjects. How many came from each of the possible linguistic backgrounds? How did they differ in language exposure? This would be informative to evaluate the generalizability of the conclusions.

Thank you for raising this point. Given the complexity of participants’ language experience, ranging between monolingual to speaking 7 different languages, we opted for a fully parametric approach in quantifying it. We used the Shannon’s entropy and Rao’s quadratic entropy equations to create continuous measures of language experience, without the constraints of a minimum sample size per language and the need to exclude participants with underrepresented languages. To add further details in our description of the language background, we summarize the language background of both samples in the newly added Table 1 presenting a breakdown of participants by number of languages they spoke, and Supplementary Table S1 listing all languages spoken by each participant.

Only the result from the multiple transverse temporal gyri (2nd TTG) is analyzed in the replicated dataset. Only the association in the right hemisphere 2nd TTG is replicated but this is not reflected in the discussion or the conclusions. The positive correlation in the right TTG is thus not attempted to be replicated.

Thank you for bringing this point to our attention. Since only few participants presented single gyri in the left (n = 7) and the right hemisphere (n = 14), the replication analysis focused on the second TTG results only. We have now commented on this fact in Section 3.5 (lines 413-415), as well as in the Discussion (lines 594-596).

The replication dataset differed in more ways than the more frequent combination of English and German experience, as mentioned in the discussion. Specifically, the fraction of monolinguals was higher in the replication dataset and the samples came from different scanners. It would be better if the primary and replication datasets were more equally matched.

Indeed, the replication sample did not fully mimic the characteristics of the main sample and a better match between the two samples would have been preferable. As elaborated in the Introduction, however, the data was split into two groups according to the date of data acquisition, which also coincided with the field strength of the scanners used for data acquisition: the first, main sample’s data were acquired on a 1.5T, the replication sample’s on 3T. We opted for keeping this split and not introducing additional noise in the analysis by using data from different field strengths at the cost of not fully matching the two datasets. Observing the established effects (even partially) in this somewhat different replication sample, however, seems in our view to further strengthen our results.

Even if the language experience and typological distance measures are a step in the right direction for correctly associating language exposure with cortical plasticity, it still is a measure that is insensitive to the intensity of the exposure. The consequences of this are not discussed.

Indeed, we agree with the reviewer that there is still a lot of grounds to cover to fully understand the relationship between language experience and cortical plasticity. We have added a paragraph to the Discussion (lines 587-592 of the revised manuscript) to bring attention to this issue:

“Future research should also further increase the degree of detail in describing the multilingual language experience, as both AoA and proficiency (used here) are not sensitive to other aspects of multilingualism, such as intensity of the exposure to the different languages, or quantity and quality of language input. Since these aspects have been convincingly shown to be associated with neural changes (e.g., Romeo, 2019), incorporating further, more detailed measures describing individuals’ language experience could further enhance our understanding of cortical plasticity in general, and how the brain accommodates variable language experience in particular.”

**Reviewer #3 (Public Review):**
Summary:The study uses structural MRI to identify how the number, degree of experience, and phonemic diversity of language(s) that a speaker knows can influence the thickness of different sub-segments of the auditory cortex. In both a primary and replication sample of adult speakers, the authors find key differences in cortical thickness within specific subregions of the cortex due to either the age at which languages are acquired (degree of experience), or the diversity of the phoneme inventories carried by that/those language(s) (breadth of experience).Strengths:The results are first and foremost quite fascinating and I do think they make a compelling case for the different ways in which linguistic experience shapes the auditory cortex.The study uses a number of different measures to quantify linguistic experience, related to how many languages a person knows (taking into account the age at which each was learned) as well as the diversity of the phoneme inventories contained within those languages. The primary sample is moderately large for a study that focuses on brainbehaviour relationships; a somewhat smaller replication sample is also deployed in order to test the generality of the effects.Analytic approaches benefit from the careful use of brain segmentation techniques that nicely capture key landmarks and account for vagaries in the structure of STG that can vary across individuals (e.g., the number of transverse temporal gyri varies from 1-4 across individuals).Weaknesses:The specificity of these effects is interesting; some effects really do appear to be localized to the left hemisphere and specific subregions of the auditory cortex e.g., TTG. However because analyses only focus on auditory regions along the STG and MTG, one could be led to the conclusion that these are the only brain regions for which such effects will occur. The hypothesis is that these are specifically auditory effects, but that does make a clear prediction that nonauditory regions should not show the same sort of variability. I recognize that expanding the search space will inflate type-1 errors to a point where maybe it's impossible to know what effects are genuine. And the fine-grained nature of the effects suggests a coarse analysis of other cortical regions is likely to fail. So I don't know the right answer here. Only that I tend to wonder if some control region(s) might have been useful for understanding whether such effects truly are limited to the auditory cortex. Otherwise one might argue these are epiphenomenal or some hidden factor unrelated to auditory experience predicting that we'd also see them in the non-auditory cortex as well, either within or outside the brain's speech network(s).

Thank you for raising this important issue. Our primary analyses indeed focused on the auditory regions, given their involvement in speech and language processing at different levels of processing hierarchy (from low – HG, to high – STG and STS). Here, we included a fairly broad range of ROIs (8 per hemisphere, 16 in total) and it has to be noted that it was only the bilateral planum temporale which showed an association with multilingualism. In the original submission we had indeed attempted at confirming the specificity of this result by performing a whole-brain vertex-wise analysis in freesurfer (see Table 3, Section 3.2, Figure S5), which again showed that the only cluster of vertices related to participants’ language experience at p < .0001 (uncorrected) was located in the superior aspect of the left STG, corresponding to the location of planum temporale and the second TTG. Lowering the threshold of statistical significance to p < .001 (uncorrected) results in further clusters of vertices whose thickness was positively associated with the degree of multilingual language experience localized in:

• Left hemisphere: central sulcus (S_cenral), long insular gyrus and central sulcus of the insula (G_Ins_lg_and_S_cent_ins), lingual gyrus (G_oc-temp_med-Lingual), planum temporale of the superior temporal gyrus (G_temp_sup-Plan_tempo), short insular gyri (G_insular_short), middle temporal gyrus (G_temporal_middle), and planum polare of the superior temporal gyrus (G_temp_sup-Plan_polar)

• Right hemisphere: angular gyrus (G_pariet_inf-Angular), superior temporal sulcus (S_temporal_sup), middle-posterior part of the cingulate gyrus and sulcus (G_and_S_cingul-Mid-Post), marginal branch of the cingulate sulcus (S_cingul-Marginalis), parieto-occipital sulcus (S_parieto_occipital), parahippocampal gyrus (G_oc-temp_med-Parahip), Inferior temporal gyrus (G_temporal_inf)

We present the result of this analysis in Author response image 1, where clusters are labelled according to the Destrieux anatomical atlas implemented in FreeSurfer:

**Author response image 1. sa4fig1:** 

As the reviewer points out, establishing relationships between our dependent and independent variables at a lower threshold of statistical significance might not reflect a true effect, and it is statistically more probable that multilingualism-related cortical thickness effects seem to be specific to the auditory regions. We do not exclude that an analysis of other pre-defined ROIs, performed at a similar level of detail as our present investigation, would uncover further significant associations between multilingual language experience and brain anatomy, but such an investigation is beyond the scope of the present work.

The reason(s) why we might find a link between cortical thickness and experience is not fully discussed. The introduction doesn't really mention why we'd expect cortical thickness to be correlated (positively or negatively) with speech experience. There is some discussion of it in the Discussion section as it relates to the Pliatsikas' Dynamic Restructuring Model, though I think that model only directly predicts thinning as a function of experience (here, negative correlations). It might have less to say about observed positive correlations e.g., HG in the right hemisphere. In any case, I do think that it's interesting to find some relationship between brain morphology and experience but clearer explanations for why these occur could help, and especially some mention of it in the intro so readers are clearer on why cortical thickness is a useful measure.

We have expanded the section of the Introduction introducing cortical thickness pointing to different microstructural changes previously associated with environmental enrichment and skill learning (lines 68-73), and hope the link between cortical thickness and multilingual language experience is clearer now:

“Such environmental effect on cortical thickness might in turn be tied to microstructural changes to the underlying brain tissue, such as modifications in dendritic length and branching, synaptogenesis or synaptic pruning, growth of capillaries and glia, all previously tied to some kind of environmental enrichment and/or skill learning (see Lövdén et al., 2013; Zatorre et al., 2012 for overviews). Increased cortical thickness may reflect synaptogenesis and dendritic growth, while cortical thinning observed with MRI may be a result of increased myelination (Natu et al., 2019) or synaptic pruning.”

In addition, we have also expanded the Discussion section providing more reasoning for the links between cortical thickness and multilingual language experience (lines 557-566):

“Experience-induced pruning is essential for maintaining an efficient and adaptive neural network. It reinforces relevant neural circuits for faster more efficient information processing, while diminishing those that are less active, or less beneficial. The cortical specialization may need to arise because phonologically more diverse language experience requires that the mapping of acoustic signal to sound categories is denser, more detailed and more intricate. As a result, the brain may need to engage in more intensive processing to discriminate between and accurately perceive the sound categories of each language. This increased cognitive demand may, in turn, require the auditory and language processing regions of the brain to adapt and become more efficient. Over time, this heightened effort for successful speech perception and sound discrimination may lead to neural plasticity, resulting in cortical specialization. This means that cortical areas become more finely tuned and specialized for processing the unique phonological features of language(s) spoken by individuals.”

One pitfall of quantifying phoneme overlap across languages is that what we might call a single 'phoneme', shared across languages, will, in reality, be realized differently across them. For instance, English and French may be argued to both use the vowel /u/ although it's realized differently in English vs. French (it's often fronted and diphthongized in many English speaker groups). Maybe the phonetic dictionaries used in this study capture this using a close phonetic transcription, but it's hard to tell; I suspect they don't, and in that case, the diversity measures would be an underestimate of the actual number of unique phonemes that a listener needs to maintain.

The PHOIBLE database uses transcription that reflects phonological descriptive data as closely as possible, according to the available descriptive sources. Different realizations of sounds are (as much as possible) marked in the database. For example, the open front unrounded vowel /a/ is listed as e.g., [a] or [a+], with the “+” sign denoting a fronted realization. This is done in PHOIBLE by the use of diacritics (see https://phoible.org/conventions) which further specify variations on the language-specific realizations of the phonemes listed in the database. Further details are available in Moran (2012). In our calculation of phoneme-based distances a sign with and without a diacritic were treated as different phonemes, and therefore the different realizations were accounted for.

That said, we fully agree with the reviewer that in fact any diversity measure will be an underestimation of the actual variation, as between-speaker micro-variation can never be fully reflected in largescale typological databases as the one used in the present study. To the best of our knowledge, however, PHOIBLE offers the most comprehensive way of allowing for quantifying cross-linguistic variation to date, and we are looking forward for the field to offer further tools capturing the linguistic variability at an ever-finer level of detail.

Discussion of potential genetic differences underlying the findings is interesting. One additional data point here is a study finding a relationship between the number of repeats of the READ1 (a factor of the DCDC2 gene) in populations of speakers, and the phoneme inventory of language(s) predominant in that population (DeMille, M. M., Tang, K., Mehta, C. M., Geissler, C., Malins, J. G., Powers, N. R., ... & Gruen, J. R. (2018). Worldwide distribution of the DCDC2 READ1 regulatory element and its relationship with phoneme variation across languages. Proceedings of the National Academy of Sciences, 115(19), 4951-4956.) Admittedly, that paper makes no claim about the cortical expression of that regulatory factor under study, and so more work needs to be done on whether this has any bearing at all on the auditory cortex. But it does represent one alternative account that does not have to do with plasticity/experience.

We thank the reviewer for bringing this important line of research to our attention, which we now included in the Discussion (lines 494-498 of the revised manuscript).

The replication sample is useful and a great idea. It does however feature roughly half the number of participants meaning statistical power is weaker. Using information from the first sample, the authors might wish to do a post-hoc power analysis that shows the minimum sample size needed to replicate their effect; given small effects in some cases, we might not be surprised that the replication was only partial. I don't think this is a deal breaker as much as it's a way to better understand whether the failure to replicate is an issue of power versus fragile effects.

Thank you for the suggestion. Indeed, the effect sizes established in the analyses using the main sample were small (e.g., f2 = 0.07). According to a power analysis performed with G*Power 3.1 (Faul et al., 2009), detecting an effect of this magnitude of the predictor of interest at alpha = .05 (two-tailed), in a linear multiple regression model with 4 predictors (i.e., 3 covariates of no-interest: sex, age, hemispheric thickness, and 1 predictor of interest), a sample of N = 114 is required to achieve 80% of power. Our partial lack of replicating the effect might therefore indeed be related to a lower power of the replication sample, rather than the effect itself being fragile.

**Recommendations for the authors:**

**Reviewer #1 (Recommendations for the Authors):**
A few remaining details that I think you can handle:(1) Was there any correction for multiple comparisons, especially when multiple anatomical measures were investigated in separate models? (e.g. ln 130).

Since three different anatomical measures were investigated in Analysis 1 and Analysis 2 (see Table 1), the alpha level of the two linear mixed models was lowered to α = .0166. Note that the p-values of the predictors of interest were p = .012 (mixed model with all auditory regions) and p = .005 (mixed model with all identified TTGs).

(2) In Table 2, since your sample skews heavily female, it would be more useful to present the counts of Male/Female totals for 1, 2, 3, 4, etc TTGs as proportions of the total for that sex rather than counts, so that the distribution across sex is more obvious.

Thank you for bringing this issue to our attention. We have now included an additional row in Table 4, with proportions of males and females presenting different total number of identified gyri in the left and the right hemisphere.

(3) (ln 161) It wasn't clear to me how you dealt statistically with the fact that some participants had only one TTG - did you simply enter "0" as a value for cortical thickness for 2, 3, etc. for those participants? If so, it's possible that this result could reflect the number of split/duplicated gyri rather than the thickness of those gyri.

Indeed, if non-existing gyri were coded with a value of “0” (it being the lowest possible thickness value), the results would reflect the configuration of TTGs (single vs multiple gyri) rather than a relationship between thickness and language experience.

The model was, however, fit to all available thickness values, and the gyri labels (1st, 2nd, 3rd) were modeled as a fixed factor with 3 levels. This procedure allowed us to localize the effect of language experience to a specific gyrus. The following formula was used with the lmer package in R:

thickness ~ age + sex + whole_brain_thickness + language_experience* gyrus*hemisphere + (1 | participant_id)

We observed a significant interaction between language experience and the 2nd gyrus (NB. no significant 3-way interaction between language experience, the 2nd gyrus and hemisphere pointed to the effect being bilateral). This result was then followed up with two linear models: one for the thickness values of the 2nd left and one for the 2nd right gyrus, each fit to the available data only (n = 130 for the left hemisphere; n = 96 for the right), see Table 5. This procedure ensured that only the available cortical thickness data were considered when establishing their relationship with our independent variable (language experience).

(4) I think more could be done in the results section to distinguish your three phonological measures--these details are evident in the Methods section, but if readers consume this paper front to back they may find it difficult to figure out what each measure really means.

Thank you. We have added more explicit list of indices used in the Introduction (lines 104-107) and in Section 2.4. As per Reviewer #2 comments, the Methods section was also moved before the Results section, hopefully further enhancing the readability of the paper.

Typos:ln 270: "weighed"--could you have meant "weighted"?

Corrected, thank you!

ln 377: "Apart from phoneme-based typological distance measure explaining"  "Apart from *the* phonemebased..."

Corrected, thank you!

**Reviewer #2 (Recommendations for the Authors):**
The interpretation of the results would be much helped by the methods section being moved to precede it. Now, much of the results section is methods summaries that would not have been needed if the reader had been presented with the methods beforehand. This is especially true for the measures of language experience and typological distances used.

Thank you. We have moved the Materials and Methods section before the Results section.

The equation in section "4.2 Language experience" should be H = - sum(p_i log2 (p_i)) and not H = - sum(p_i log2(i)).

Corrected, thank you!

It is unclear what "S" represents in the equation in the section "4.4 Combining typology and language experience (indexed by AoA)".

The explanation has been added, thank you!